# Discerning Decision-Making Process of Deep Neural Networks with Hierarchical Voting Transformation

**Ying Sun**[1,2,5,†], **Hengshu Zhu**[2,*], **Chuan Qin**[2], **Fuzhen Zhuang**[3,6*], **Qing He**[1,5], **Hui Xiong**[4]

[1] IIP, Institute of Computing Technology, Chinese Academy of Sciences
[2] Baidu Talent Intelligence Center, Baidu Inc.
[3] Institute of Artificial Intelligence, Beihang University
[4] Artificial Intelligence Thrust, The Hong Kong University of Science and Technology
[5] University of Chinese Academy of Sciences
[6] SKLSDE, School of Computer Science, Beihang University
{sunying17g, heqing}@ict.ac.cn, {zhuhengshu, chuanqin0426}@gmail.com,
zhuangfuzhen@buaa.edu.cn, xionghui@ust.hk

## Abstract

Neural network based deep learning techniques have shown great success for numerous applications. While it is expected to understand their intrinsic decision-making processes, these deep neural networks often work in a black-box way. To this end, in this paper, we aim to discern the decision-making processes of neural networks through a hierarchical voting strategy by developing an explainable deep learning model, namely Voting Transformation-based Explainable Neural Network (VOTEN). Specifically, instead of relying on massive feature combinations, VOTEN creatively models expressive single-valued voting functions between explicitly modeled latent concepts to achieve high fitting ability. Along this line, we first theoretically analyze the major components of VOTEN and prove the relationship and advantages of VOTEN compared with Multi-Layer Perceptron (MLP), the basic structure of deep neural networks. Moreover, we design efficient algorithms to improve the model usability by explicitly showing the decision processes of VOTEN. Finally, extensive experiments on multiple real-world datasets clearly validate the performances and explainability of VOTEN.

## 1 Introduction

Neural network based deep learning techniques have attracted great attention from both academia and industry in the past decade. Compared with classic machine learning models, deep neural networks have much higher expressiveness and adaptability for complicated data input, and thus have made tremendous success in various application domains, such as Computational Vision [29, 56], Natural Language Processing [54, 20], and Recommender Systems [32, 24]. Nevertheless, since deep neural networks usually have complicated connections of hidden units, a long-standing challenge is how to decipher what's inside the black box of models for understanding their intrinsic decision-making processes. Indeed, in many real-world scenarios, such as business analysis [33, 57, 49] and human resource management [50, 44, 46, 44], the lack of model explainability makes people less likely to be convinced when the decision-making process of the model is not understandable. This prevents a broader application of deep neural networks.

While many research efforts have been made in developing explainable deep learning models [45, 10], most of existing works focus on post-hoc explanation [5, 18], i.e., designing metrics to measure the feature relevance/contribution to the outputs of a trained model. Although these methods have

---

*Corresponding authors.
†This work was accomplished when Ying Sun was an intern in Baidu supervised by Hengshu Zhu.

35th Conference on Neural Information Processing Systems (NeurIPS 2021).

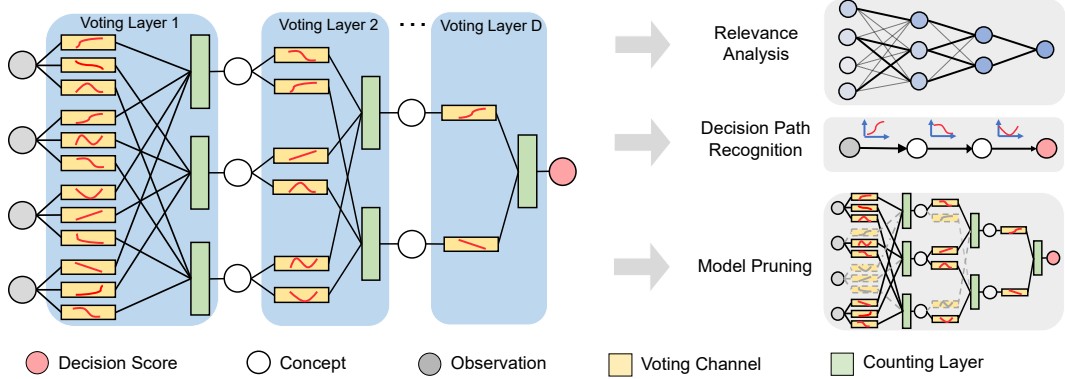

Figure 1: Structure overview of VOTEN.

made progress on finding important features, the decision-making process of deep learning models is still not available for users. For example, users often need to guess the reason why a prediction is made from the relevant features instead of understanding the decision-making process. Indeed, understanding the intrinsic decision-making process of deep neural networks is a non-trivial task. A major reason is that deep neural networks usually involve massive feature combinations to gain expressiveness on fitting complicated functions. During this process, the effect of features and hidden units may be largely coupled with each other. This indicates that the decision logic of models is inherently buried in the massive feature combinations. Meanwhile, it is difficult for human to understand the intrinsic modeling process of deep learning in a natural manner. Therefore, a key point on improving the explainability of deep neural networks is to decouple the feature combinations and make the modeling process consistent with human decision process. In this way, the model will have an explicit decision-making process and become human-understandable.

To this end, in this paper, we propose an explainable deep learning model, namely Voting Transformation-based Explainable Neural Network (VOTEN). Specifically, VOTEN assumes the transformation from the input to the output to be a hierarchical voting process. During this process, lower-level concepts vote for higher-level concepts layer-by-layer in an expressive but explainable way. Instead of relying on massive feature combinations, VOTEN creatively models expressive single-valued voting functions between explicitly modeled hidden concepts to gain expressiveness on fitting complicated functions. This process is explainable for its consistency with human decision-making process. We first theoretically analyze the major components of VOTEN and prove the relationship and advantages of VOTEN compared with Multi-Layer Perceptron (MLP), which is the basic structure of deep neural networks. The results show that MLP can be derived from VOTEN by using the inexpressive voting functions. Accordingly, we further analyze the effect of inherent votings and design efficient algorithms for pruning and explaining VOTEN. Finally, we evaluated VOTEN on multiple real-world datasets with comprehensive experiments. The experimental results demonstrate that VOTEN generally promotes powerful and explainable deep learning. Specifically, VOTEN (1) significantly raises prediction performance; (2) exponentially decreases feature combinations; and (3) supports efficient pruning and effective feature analysis. Meanwhile, we also visualize the intrinsic decision-making process of VOTEN through case studies, which show VOTEN is explainable and can discover meaningful latent concepts.

## 2 VOTEN

In this section, we introduce the technical details of VOTEN.

### 2.1 Structure

When dealing with complicated information, we usually aggregate them step-by-step to form complicated inference. Although deep neural networks also extract abstract features layer-by-layer, they are still difficult to be explained. A major difference is that we can induce explicit concepts, which makes us able to explain our inference in a straightforward way.

For example, we can predict that "This student may get high GPA because he/she works very hard". If the more explicit explanation is needed, we can further explain that "I observed that he/she stay long in the library". In this process, with the observation that "stay long in the library", we first

infer "hard-working", then infer "high GPA". Moreover, when trying to make more comprehensive inference, we may get more observations for existing concepts (e.g., "little missing of classes" may also imply "hard-working"), or induce more concepts (e.g., "learning-efficiency"). This kind of process is straightforward and conforms with our way of understanding. However, deep neural networks cannot model concepts in an explicit way. Instead, concept information are embedded in massive hidden units. As we cannot understand these induced concepts, neural networks are like unified, inseparable complicated functions, even though they are actually performing information aggregation. Along this line, we believe a natural way to understand how a model decides the output is to make the intrinsic concepts explicit for human.

Therefore, we propose VOTEN, whose structure overview is shown in Figure 1. VOTEN hierarchically models a small number of explicit concepts. During inference, the higher-level concepts will be induced via information aggregation from the lower-level concepts. To achieve meaningful information aggregation, VOTEN focuses on quantifying relationship between individual concepts of different levels. For better understanding, in VOTEN, observations and concepts can be regarded as *voters*. Based on their own value, each voter independently votes for the higher-level concepts. The votes are aggregated to get the value estimation of each higher-level concept. These concepts will further vote for the next level. In the training process, the model learns to builds intermediate concepts and their quantitative voting functions.

Formally, for a VOTEN model with $D$ levels of concepts, we use $\mathcal{C}_i^d$ to denote the $i$-$th$ concept in the $d$-$th$ layer, where $\mathcal{C}_i^0$ denotes an input feature. We refer to transformations between adjacent levels of concepts as a *voting layer*. In the $d$-$th$ voting layer, each concept $\mathcal{C}_i^d$ votes for each higher-level concept $\mathcal{C}_j^{d+1}$ with an independent *voting channel* $\mathcal{V}_{i,j}^d$. $\mathcal{V}_{i,j}^d$ takes the value of $\mathcal{C}_i^d$ as the input and votes with a single-valued nonlinear function $f_{i,j}^d : \mathbb{R} \to \mathbb{R}$. Then, a counting layer gets weighted average of the votes and estimates the value of $\mathcal{C}_j^{d+1}$ as

$$x_j^{d+1} = \sum_{k=1}^{n_d} a_{k,j}^d f_{k,j}^d(x_k^d) \qquad \text{s.t. } \forall d, j, \ \sum_{k=1}^{n_d} a_{k,j}^d = 1, \tag{1}$$

where $n_d$ denotes the number of concepts in the $d$-$th$ layer, $x_k^d$ denotes the value of $\mathcal{C}_k^d$, $a_{k,j}^d$ denotes the weight of $\mathcal{V}_{i,j}^d$. The concepts of the last layer is regarded as the decision score, which generates the model output with $\boldsymbol{o} = F^{VOTEN}(\boldsymbol{x}^D)$. In particular, to assure the ability of the voting functions on effective concept transformation, VOTEN models each voting function with a voting network. The voting network can be designed in complicated ways without influencing model explainability, as long as the function is still single-valued. For example, we can adopt weight-sharing structures to reduce model complexity.

## 2.2 Why is VOTEN more explainable than MLP?

In this part, we theoretically discuss VOTEN's advantages over MLP. Specifically, we claim that voting expressiveness is the core proposition of VOTEN that raises explainability.

**Theorem 1** *MLP can be derived from a subset of degenerated VOTEN models whose voting functions in the form of $f_{i,j}^d(x_i^d) = w_{i,j}^d \sigma(x_i^d) + b_{i,j}^d$, where $\sigma$ is a predefined activation function, the scalars $w_{i,j}^d, b_{i,j}^d \in \mathbb{R}$ are trainable parameters.*

*Proof.* Please refer to Appendix A.

From this point of view, MLP also conforms to human inference. However, MLP is still difficult to explain. Indeed, we can more easily understand an inference process with (1) fewer concepts, (2) shorter concept transformations, and (3) fewer reasoning patterns. Under VOTEN schema, we show how inexpressive voting channels make MLP disobey these three traits.

**Corollary 1** *In MLP, votings from the same concept are linearly correlated.*

This means individual voters in MLP are weak in distinguishing different concepts in the next level. Therefore, MLP relies on highly complicated feature combinations of a large number of deeply tiled voters to achieve high fitting ability. During this process, necessary intermediate information is inherently modeled through combinatorial effects of hidden concepts with inexplicit meanings. Indeed, previous works have proved that human-understandable concepts are inherently mounted in the hidden units of neural network models [28].

**Corollary 2** *In MLP, the hypothesis space for voting distribution is limited to scaling and shifting an input distribution.*

This means that MLP voters cannot always induce complicated concepts that have different distribution from their value. As a result, the input needs to go through a long path of transformations between similar concepts until it contributes to the output. Moreover, the deeply tiled massive concepts make each input feature has massive paths to the output, which brings a large number of possible decision-making patterns of the model. In addition, the effect of votings can easily couple and cancel each other out in the downstream calculations. As a result, separately analyzing the role of individual concepts or decision paths becomes meaningless.

Different from MLP, VOTEN has far more expressive voting functions. It directly models nonlinear transformations between essential intermediate concepts without relying on massive feature combinations and naive transformations. As a result, VOTEN's decision-making process is explicit with only a small number of meaningful concepts, thus is explainable.

### 2.3 Explaining the effect of votings

In VOTEN, individual voting channels play explicit roles in influencing the model decision. In this part, we analyze the effect of votings. First, we use a concept function $g_i^d : \mathbb{R}^{n_0} \to \mathbb{R}$ to represent the transformation from model inputs $\boldsymbol{x}^0 \in \mathbb{R}^{n_0}$ to $\mathcal{C}_i^d$, which decides the meaning of the concept.

**Definition 1** *In VOTEN, we refer to two concept functions $g$ and $g'$ as equivalent iff. there exists an invertible function $\Phi$, so that $\forall \boldsymbol{x}^0 \in \mathbb{R}^{n_0}$, $\Phi(g(\boldsymbol{x}^0)) = g'(\boldsymbol{x}^0)$.*

Since $\Phi$ is invertible, the outputs of two equivalent concept functions have one-to-one correspondence over all the possible inputs. Then, they can effect equally in the decision-making process.

**Theorem 2** *In VOTEN, if replacing a concept function with an equivalent form, there exists a way to replace its voting functions so that all the downstream concept functions stay unchanged.*

*Proof.* Please refer to Appendix A.

To analyze how voting channels effect on concept functions, we can reformulate Equation 1 as

$$x_j^{d+1} = g_j^{d+1}(\boldsymbol{x}^0) = \sum_{i=1}^{n_d} a_{i,j}^d (f_{i,j}^d(g_i^d(\boldsymbol{x}^0)) - \mathbb{E}_{\boldsymbol{x}}[f_{i,j}^d(g_i^d(\boldsymbol{x}))]) + b_j^{d+1}, \tag{2}$$

where $\mathbb{E}_{\boldsymbol{x}}[f_{i,j}^d(g_i^d(\boldsymbol{x}))]$ denotes the expectation of $f_{i,j}^d$ over all the instances, $b_j^{d+1}$ is a sample-independent bias. In particular,

$$b_j^{d+1} = \sum_{i=1}^{n_d} a_{i,j}^d \mathbb{E}_{\boldsymbol{x}}[f_{i,j}^d(g_i^d(\boldsymbol{x}))] = \mathbb{E}_{\boldsymbol{x}}[g_j^{d+1}(\boldsymbol{x})]. \tag{3}$$

For simplicity, we use $\overline{h_{i,j}^d}$ to denote $\mathbb{E}_{\boldsymbol{x}}[f_{i,j}^d(g_i^d(\boldsymbol{x}))]$. Notably, by adding an arbitrary bias to $g_j^{d+1}$, we obtain an equivalence of the original concept. According to Theorem 2, we can construct a model with exactly the same expression (i.e., equivalent concepts and the same predictions) as the previous one. This implies that VOTEN may converge to models with differed internal bias but exactly the same decision-making process, indicating VOTEN explanation should be independent of concept bias. According to Equation 2 and 3, voting channels' average only influence concept bias while the voting deviation $f_{i,j}^d(g_i^d(\boldsymbol{x}^0)) - \overline{h_{i,j}^d}$ reveals the effect of $\mathcal{V}_{i,j}^d$ for the decision. This can be intuitively explained as each voter can vote with different basic scores and only the deviation from the basic score reflects their judgement for a specific instance. Similarly, from the global view, we reformulate the concept function as $x_j^{d+1} = \sum_i a_{i,j}^d \sigma_{i,j}^d K_{i,j}^d(\boldsymbol{x}^0) + b_j^{d+1}$, where $K_{i,j}^d(\boldsymbol{x}^0) = \frac{f_{i,j}^d(g_i^d(\boldsymbol{x}^0)) - \overline{h_{i,j}^d}}{\sigma_{i,j}^d}$.

Notably, $K_{i,j}^d(\cdot)$ generates a distribution with mean 0 and variance 1 over all the instances. Therefore, $a_{i,j}^d$ and $\sigma_{i,j}^d$ jointly decide the overall effect of votings. Specifically, the counting layer explicitly adjusts $a_{i,j}^d$ so that reliable voters have stronger influences. Meanwhile, the voter implicitly adjusts $\sigma_{i,j}^d$. When the concept is less related to the target concept and cannot support proper votes, they decrease $\sigma_{i,j}^d$ and tend to always vote the basic score to avoid disturbing the model. Otherwise, they increase $\sigma_{i,j}^d$ and vote confidently to lead the model to correct estimation.

Based on the above analysis, we can easily design algorithms to ease both VOTEN local and global explanation, such as recognizing decision paths and quantifying the concept/feature relevance to the prediction, which can be found in Appendix B and Appendix C.

Table 1: Model Performance. We conducted 10 independent runs on each dataset and show the average $\pm$ standard deviation of AUC and AP. In particular, for multi-classifications, we estimated the macro average of each metric. We also did significance test, where * and ** denote significantly (i.e., $p$-value $\leq 0.05$) and very significantly (i.e., $p$-value $\leq 0.01$) worse than VOTEN.

|     |      | DT [43] | RF [48] | LGB [27] | MLP [22] | NAM [7] | VOTEN |
|-----|------|---------|---------|----------|----------|---------|-------|
| MR  | PR   | 0.2557** | 0.4624** | 0.4817** | 0.4870** | 0.4412** | **0.5007** |
|     | $\pm$ | 0.0007 | 0.0010 | 0.0007 | 0.0040 | 0.0014 | 0.0041 |
|     | AUC  | 0.6762** | 0.9007** | 0.9209** | 0.9199** | 0.9031** | **0.9237** |
|     | $\pm$ | 0.0096 | 0.0006 | 0.0001 | 0.0008 | 0.0014 | 0.0005 |
| RP  | PR   | 0.0177** | 0.0349** | 0.0525* | 0.0516** | 0.0485** | **0.0550** |
|     | $\pm$ | 0.0001 | 0.0011 | 0.0017 | 0.0014 | 0.0014 | 0.0001 |
|     | AUC  | 0.5134** | 0.6566** | 0.7305* | 0.7250** | 0.7102** | **0.7322** |
|     | $\pm$ | 0.0014 | 0.0052 | 0.0016 | 0.0017 | 0.0082 | 0.0003 |
| CT  | PR   | 0.8119** | 0.9763** | 0.9753** | 0.9662** | 0.7000** | **0.9783** |
|     | $\pm$ | 0.0010 | 0.0003 | 0.0003 | 0.0018 | 0.0008 | 0.0013 |
|     | AUC  | 0.9396** | 0.9979** | 0.9965** | 0.9965** | 0.9497** | **0.9983** |
|     | $\pm$ | 0.0005 | 0.0001 | 0.0001 | 0.0002 | 0.0001 | 0.0001 |
| CI  | PR   | 0.2514** | 0.6348** | **0.6972** | 0.6215** | 0.6567** | 0.6522 |
|     | $\pm$ | 0.0022 | 0.0014 | 0.0001 | 0.0030 | 0.0043 | 0.0019 |
|     | AUC  | 0.7250** | 0.9388** | **0.9566** | 0.9454** | 0.9506** | 0.9508 |
|     | $\pm$ | 0.0018 | 0.0005 | 0.0001 | 0.0004 | 0.0006 | 0.0002 |
| HG  | PR   | 0.6408** | 0.8505** | 0.8590** | 0.8297** | 0.7897** | **0.8612** |
|     | $\pm$ | 0.0002 | 0.0001 | 0.0001 | 0.0007 | 0.0003 | 0.0005 |
|     | AUC  | 0.6705** | 0.8429** | 0.8459** | 0.8157** | 0.7751** | **0.8481** |
|     | $\pm$ | 0.0002 | 0.0001 | 0.0001 | 0.0007 | 0.0001 | 0.0005 |
| AS  | PR   | 0.0081** | 0.0085** | 0.0120 | 0.0113** | 0.0108** | **0.0120** |
|     | $\pm$ | 0.0001 | 0.0001 | 0.0001 | 0.0001 | 0.0001 | 0.0001 |
|     | AUC  | 0.4999** | 0.5139** | **0.6026** | 0.5917** | 0.5742** | 0.5975 |
|     | $\pm$ | 0.0002 | 0.0009 | 0.0001 | 0.0005 | 0.0018 | 0.0021 |

## 2.4 VOTEN supports effective model pruning

In VOTEN, concepts are estimated by averaging the votes. This means we can delete a voting channel while keeping the physical meaning of the target concept unchanged. This supports effective link pruning. Specifically, since only the voting deviation from the average decides the effect, we can assume the absent channel votes the basic score regardless of the input. Formally, the value of $\mathcal{C}_j^{d+1}$ is estimated as $\hat{x}_j^{d+1} = \sum_{i=1}^{n_d} I_{i,j}^d a_{i,j}^d (f_{i,j}^d(x_i^d) - \overline{h_{i,j}^d}) + b_j^{d+1}$, where $I_{i,j}^d \in \{0,1\}$ indicates if $\mathcal{V}_{i,j}^d$ is not absent. Furthermore, since only involving a small number of voting channels, we can achieve network pruning on VOTEN by exhaustively exploring how the performance will get influenced if some voting channels are absent, which is infeasible for MLP. In MLP, we usually prune unimportant hidden units to reduce model complexity. However, usually not all the voting channels from an important concept are necessary. In VOTEN, these unnecessary concept transformations can be further eliminated to not only reduce model complexity but also raises the explainability of the model. In Appendix D, we give an efficient VOTEN pruning algorithm with a lazy updating strategy.

# 3 Experiment

To evaluate the effectiveness and explainability of VOTEN for seizing comprehensive decision-making patterns. We conducted experiments[1] with 6 large public datasets, including Context-aware Multi-Modal Transportation Recommendation (MR) [2, 58], IJCAI-18 Search Conversion Rate Prediction (RP) [3], Forest Cover Type Prediction (CT) [12], Census-Income Prediction (CI) [38], Allstate Claim Prediction (AS) [1], and Higgs boson dataset (HG) [11]. The detailed descriptions of experimental setup can be found in Appendix E.

## 3.1 Performance Evaluation: Can VOTEN achieve higher performance than MLP?

We used two widely adopted metrics for imbalanced classification performance evaluation, including Area Under ROC Curve (AUC) [14] and average precision (AP) [19], whose higher value means

---
[1]Our code is available at https://github.com/sunyinggilly/VOTEN

higher performance. In Table 1, we compare the performance of VOTEN with several baselines, including Decision Tree (DT) [43], Random Forest (RF) [48], LightGBM (LGB) [27], MLP [22], and Neural Additive Model (NAM) [7] (see Appendix E.2 for detail). For each dataset, we have carefully tuned the parameters of the baselines to achieve their best performance. Especially, detailed analysis on MLP parameters can be found in Appendix E.3. It can be observed that, while deep neural networks are powerful when incorporated with purposely designed modules or prior knowledge for specific tasks, its standard form (i.e., MLP) without task-specific structures may perform worse than LightGBM, which have also been shown by many previous studies [47, 8, 4]. Indeed, LightGBM is believed to be powerful in handling structured data and often appears as the major model of the top solutions in data-mining competitions [58, 35]. In contrast, VOTEN, also in its standard form, significantly outperforms MLP for all these tasks and comparable with LightGBM. Indeed, on many datasets, it outperforms LightGBM if complicated feature engineering is not performed. This shows the effectiveness of VOTEN in terms of handling real-world problems. Indeed, VOTEN is more suitable for handling data-mining tasks since it has more reasonable decision-making process. It should be noticed that, similar to MLP, VOTEN is a standard and generic model that can be easily expanded to task-specific networks to raise model performance (for example, we can replace MLP with VOTEN in DeepFM [24]). While this paper focuses on the generic performance of standard VOTEN, it shows the possibility of building more powerful deep learning solutions for a wide range of applications.

## 3.2 Explanation Complexity: Is the decision-making process of VOTEN recognizable?

In Table 2, we compare the explanation complexity of MLP and VOTEN. As we have discussed in Section 2.2, we use the number of feature combinations, the length of decision paths, and the number of possible decision-making patterns to show the explanation complexity of a model. In particular, we trained two VOTEN models with different settings for each dataset. Specifically, "VOTEN$^-$" is comparable to the best performance of MLP, with the least feature combinations. "VOTEN" is the one with the best performance. It can be observed that VOTEN greatly reduces feature combinations and decision paths. For example, on the MR dataset, MLP needs 7 voting layers that each contain 128 concepts. This brings an exponentially large number of long decision paths. In contrast, VOTEN achieves comparable performance with 16 hidden concepts in total. Then, each feature only has 16

Table 2: Explanation complexity. "#C/L" counts hidden concepts in each layer. "#P/F" counts possible decision paths from each feature.

| Data | Model | Performance | | #C/L | Depth | #P/F |
|------|-------|-----|-----|------|-------|------|
| | | AP | AUC | | | |
| MR | MLP | 0.487 | 0.920 | 128 | 7 | $2^{49}$ |
| | VOTEN$^-$ | 0.497 | 0.922 | 16 | 1 | 16 |
| | VOTEN | **0.501** | **0.924** | 16 | 2 | 256 |
| RP | MLP | 0.052 | 0.725 | 12 | 3 | 1,728 |
| | VOTEN$^-$ | 0.055 | 0.732 | 0 | 0 | 1 |
| | VOTEN | **0.055** | **0.732** | 8 | 2 | 64 |
| CT | MLP | 0.966 | 0.996 | 128 | 7 | $2^{49}$ |
| | VOTEN$^-$ | 0.969 | 0.997 | 32 | 2 | 1,024 |
| | VOTEN | **0.978** | **0.998** | 64 | 2 | 4,096 |
| CI | MLP | 0.622 | 0.945 | 32 | 2 | 1,024 |
| | VOTEN$^-$ | 0.652 | 0.950 | 0 | 0 | 1 |
| | VOTEN | **0.652** | **0.951** | 4 | 2 | 16 |
| HG | MLP | 0.830 | 0.816 | 64 | 4 | $2^{24}$ |
| | VOTEN$^-$ | 0.831 | 0.816 | 8 | 1 | 8 |
| | VOTEN | **0.862** | **0.848** | 64 | 2 | 4,096 |
| AS | MLP | 0.011 | 0.592 | 64 | 4 | $2^{24}$ |
| | VOTEN$^-$ | 0.012 | 0.594 | 16 | 1 | 16 |
| | VOTEN | **0.012** | **0.598** | 64 | 2 | 4,096 |

possible paths with a length of 2 to reach the output. This significantly eases the understanding of the decision-making process. For some datasets, VOTEN with no intermediate concepts (i.e., features directly vote for the prediction) can achieve comparable performance to MLP. Moreover, even when reaching its best performance, VOTEN still has a much smaller explanation complexity than MLP. It should be noticed that we have listed the maximum possible number of decision paths for convincing illustration. In practice, we can easily distinguish a fewer number of important decision paths, which will be shown in the following experiments.

## 3.3 Case Study: How to explain a VOTEN model?

We show how to explain a VOTEN model with an example in the MR dataset. Specifically, we first analyze the global decision-making process and discover the meaning of concepts by observing voting functions on important decision paths. Then, we locally explain how the inputs of an instance hierarchically vote the final prediction.

**Task Description.** The task is to recommend the transport mode for online map app users, given a user and an Origin-Destination (OD) pair. The features mainly contain user portraits and an ordered list of recommended plans of the map app. Each plan consists of transport mode, time, distance, and price. We deleted the first recommended plan's mode information since it is too strongly correlated

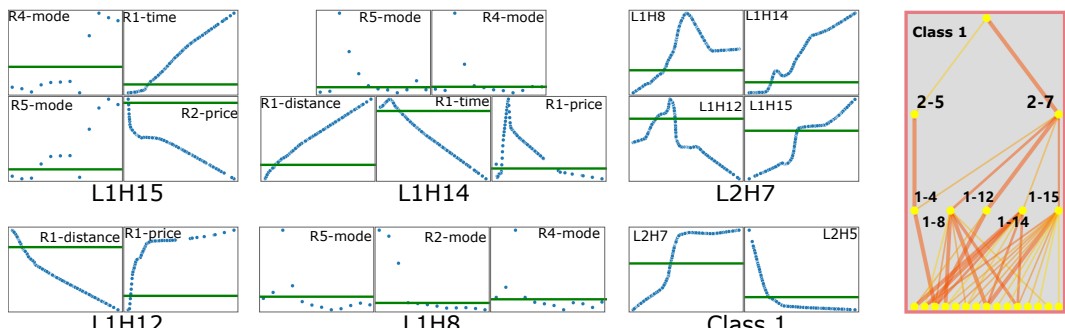

Figure 2: Case study of VOTEN global decision-making process in MR dataset. We show the important paths from inputs to one output, where wide lines indicate important voting channels. The concept L$x$H$y$ indicates the $y$-$th$ concept in the $x$-$th$ layer. We visualize the important voting functions for each concept along the paths relevant to L2H7, where blue lines show the function while green lines show the average vote.

with the label (many users choose the first recommendation as default). In this way, we can better observe how the model incorporates complicated information for meaningful decisions.

**Global Decision-Making Process.** We focus on the decision-making process for the class "Subway". First, we discover the important transformations from observations to the prediction with our decision path recognition algorithm. The results are shown in Figure 2. Next, we analyze the meaning of concepts from the bottom to the top. L1H8 gets larger when modes of more recommended plans are "Subway", which we regard as "*OD pair with flexible subway-based plans*". L1H12 decreases with higher prices and lower distances, which we regard as "*OD pair's distance-cost performance*". L1H14 observes if the OD pair is distant but still available with inexpensive and fast transportation. Besides, it also observes if modes of public transportation ever appear in the plan list. Therefore, we regard L1H14 as "*distant OD pair with convenient and economical public transportation*". L1H15 is sensitive to a time-consuming top-1 plan. It also observes the other plans' modes and prices to estimate if trading money for efficiency is infeasible. Therefore, we regard L1H15 as "*no choice but a time-consuming transportation*". In the second layer, L2H7 is estimated based on L1H8, L1H12, L1H14, and L1H15. It gets higher if many subway-based plans available (higher L1H8), transportation with balanced distance-cost is recommended (has a peak for L1H12), distant OD pair but still has convenient public transportation (higher L1H14), or costly time-saving transportation is infeasible (higher L1H15). Comprehensively considering these reasons, it implies "*OD pair suitable for subway transportation*, which votes for the score of "Subway" with a monotonic transformation. Along the other path, L1H4 is a concept "*OD pair with inexpensive transportation*". Then, L2H5 is also about the price since it is mainly based on L1H4. It should be noticed that its estimation may still be enhanced by other lower-level concepts when dealing with specific instances, although they are less important from the global view. Finally, L2H5 and L2H7 vote nonlinearly to the score so that the model can make accurate quantitative predictions. With the above analysis, we can qualitatively understand the logic of VOTEN on recommending "Subway". Actually, the decision-making process is quantitatively more complicated and can handle more special cases. In practice, domain experts can thoroughly analyze the shape and gradients of the voting functions to get more insights into the concepts. This may help researchers to find new concepts and develop new theories, especially in fields such as psychology and management, where scientists work on finding mechanisms linking observations to outcomes. Extra visualizations on global decision path and voting functions can be found in Appendix F and Appendix G.

**Local Decision-Making Process.** Then, we analyze the decision paths for specific instances. In Figure 3, we visualize the most important paths for a sample with a high score for "Subway", which are filtered with a small threshold in our local decision path recognition algorithm. Based on the global analysis, we can easily tell how the observations gradually transformed to higher-level concepts. Specifically, for L1H12, the vote from price is filtered out, showing the OD pair's price is normal over the dataset. However, it finds the OD pair to be distant, which contributes to a relatively larger *distance-cost performance* than average, given the normal price. This indicates the plan to be relatively more economical. L2H7 observes the value of L1H12, and finds that the OD pair can

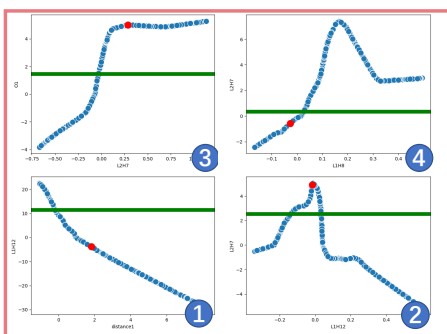 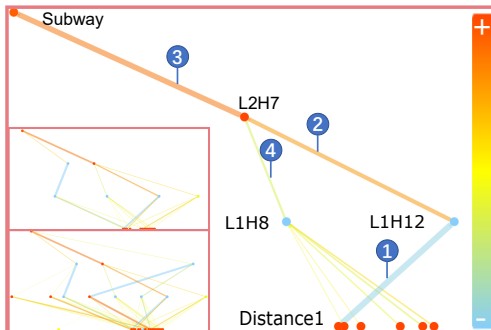

Figure 3: Case study of VOTEN local decision-making process in MR dataset, where wider and darker lines indicate stronger influence (negatively in blue and positively in red). The bottom left shows filtered paths when we gradually bring up the thresholds. We also show the important voting functions, where the green lines show the average and red points show vote for the current instance.

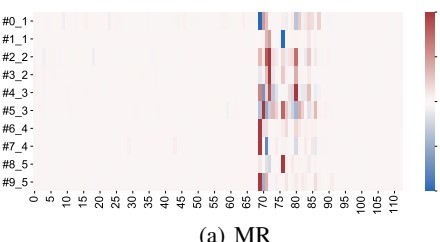 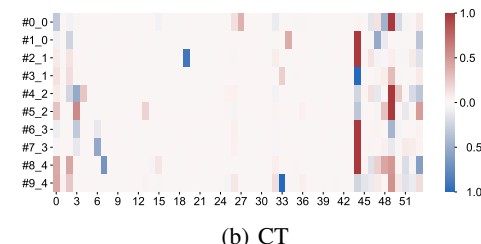

(a) MR                 (b) CT

Figure 4: Heatmap for the propagation-based relevance in VOTEN. The x-axis represents features and the y-axis represents the instance-output pairs, where $\#I\_O$ indicates the feature relevance of the $I$-th instance to class $O$. Red means positive relevance while blue means negative relevance.

choose transportation whose *distance-cost performance* more balanced than average cases (reaches the peak of voting). In this case, L2H7 votes high for subway, which is an economical and balanced transportation mode. On the other decision path, L1H8 thinks "Subway" may not be suitable since subway-based plans seldom appear in the list, thus votes negatively for L2H7. But L1H12 makes a very confident judgment based on the balance of distance-price performance, which dominates L2H7 and makes the model predict correctly. Interestingly, we find one of the most important strategies in this task is to guess the transportation mode most recommended by the app (the information that we hide in prior), which is reasonable. On the one hand, the app's recommender system trained with abundant information can naturally achieve high performance. On the other hand, many users will click the first recommendation as default. In addition to this strategy, the model will use more important decision paths to achieve more accurate predictions. As we gradually increase the threshold, more decision-making patterns appeared. Extra visualizations on local explanation of other datasets can be found in Appendix H.

### 3.4    Relevance Analysis: Can VOTEN help quantify feature relevance?

**Propagation-Based Relevance.** Motivated by relevance propagation [10], we propose an algorithm (see Appendix D) to quantify the relevance of features and concepts, based on the important decision paths. The short decision paths of VOTEN decreases error accumulation during the propagation and enables more accurate relevance estimation. In Figure 4, we visualize the propagation-based feature relevance with heatmaps. Obviously, features from #69 to #86 are generally important in the MR dataset, among which the first several features (information about the top-1 plan in the list) are the most relevant. Furthermore, the relevant features vary for different samples in terms of different classes, which indicates VOTEN to predict in multiple patterns. For example, Figure 4(b) shows that VOTEN assigns high score to class 2 for sample 4 and sample 5 with different reasons. Specifically, feature #3 and #53 contribute positively for sample 5 while negatively for sample 4. Instead, #50 is more positively relevant for sample 4. Extra visualizations can be found in Appendix I.

**Single-Sighted Prediction Strength.** We can also estimate the feature relevance from the view of model performance when the prediction is supported by a single voter in some layer. Specifically,

Table 3: Features with top-5 single-sighted prediction strength for class 0 and class 1 of MR dataset in VOTEN. "Input" indicates the AUC of ranking the samples according to the feature's value.

| Model | Class 0 | | | | | Class 1 | | | | |
|-------|-------|-------|-------|-------|-------|-------|-------|-------|-------|-------|
| | Top-1 | Top-2 | Top-3 | Top-4 | Top-5 | Top-1 | Top-2 | Top-3 | Top-4 | Top-5 |
| VOTEN | 0.675 | 0.653 | 0.647 | 0.606 | 0.592 | 0.747 | 0.632 | 0.625 | 0.587 | 0.566 |
| Input | 0.596 | 0.533 | 0.572 | 0.600 | 0.586 | 0.535 | 0.527 | 0.534 | 0.555 | 0.525 |

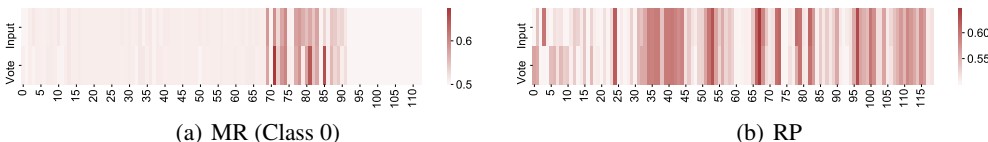

(a) MR (Class 0)          (b) RP

Figure 5: Single-sighted prediction strength in VOTEN. We also show the AUC of linearly ranking the samples as comparisons. Darker color means higher AUC.

we disable the other voters in a similar way as we do in model pruning. Table 3 shows 5 MR features achieving the highest AUC when conducting single-sighted prediction for class 0 and class 1 in VOTEN. As a comparison, we also show the AUC of directly ranking with the feature, which reveals the feature's linear correlation with the prediction. In particular, since there can be negative correlations, we evaluate AUC for the rank in increasing and decreasing order, and use the larger one as the performance. Moreover, Figure 5 shows the results on all the features. It can be observed that, even if approximated into single-sighted, VOTEN significantly raises AUC, showing VOTEN to recognize important features and strengthen their effectiveness with nonlinear transformations. Especially, while feature #85 originally seems not correlated with the output, VOTEN finds it in practice nonlinearly very relevant to the output. This proves the effectiveness of VOTEN on quantifying the inherent nonlinear relationships between the observations and the prediction. Interestingly, VOTEN weakens the effect of some features (e.g., feature #3 for RP) to prevent them from disturbing the prediction. Extra visualizations can be found in Appendix J.

## 3.5 Pruning Experiment

We conducted pruning experiments on VOTEN for MR and CT datasets with our pruning algorithm. As we gradually deleted voting channels, we monitored the change of AUC during this process, which are shown in Figure 6. For MR, AUC is still near 0.924 after pruning nearly half of the voting channels. For CT, AUC is still near 0.998 after pruning a quarter of the channels. Interestingly, proper pruning may slightly raise model performance, which is reasonable as a simple model has less chance of over-fitting. In practice, operations like fine-tuning can be adopted

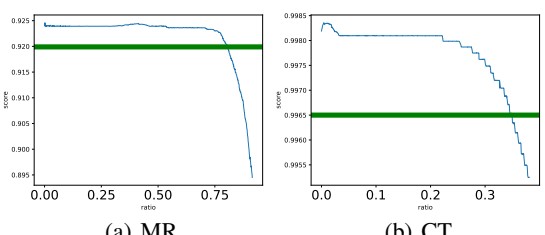

(a) MR     (b) CT

Figure 6: VOTEN pruning experiments. The x-axis shows the ratio of pruned channels while the y-axis shows AUC. The green line shows MLP's performance.

to further raise the performance of the pruned VOTEN model. Then, the model can be further compressed without affecting the prediction much. These results prove VOTEN to support effective pruning, which is helpful. We can use complicated information for training and prune the model to decrease the complexity for storage, calculation and explanation.

## 4 Related Work

**Post-Hoc Deep Learning Explanation.** Post-Hoc explaining algorithms analyze the relevance of features in a model-free way, mainly including propagation-based methods [30, 6] and perturbation-based methods [59, 56]. Propagation-based methods propagate the relevance score backward to the inputs. For example, Simonyan *et al.* [45] generates saliency maps with the gradients of the output category with respect to the inputs. Bach *et al.* [10] proposed Layer-wise Relevance Propagation (LRP), which designed effective rules for the propagation. Perturbation-based methods explain

model behavior by observing how the output reacts to purposely perturbed or constructed inputs. For example, Local Interpretable Model-Agnostic Explanations (LIME) [42] trains a local explainable approximation model around the prediction with randomly perturbed features and the corresponding outputs. SHapley Additive exPlanations (SHAP) [34] estimates the Shapley value of features to measure their contribution to model performance. In addition to features, some works estimate concept importance for a model [23, 53, 28]. For example, Kim *et al.* [28] learn the representation of human-understandable concepts with labeled concept-relevant examples and estimate concept sensitivity according to the directional derivative towards the concepts. Along this line, abundant works have been proposed to further raise the effectiveness of post-hoc interpretation algorithms [31, 55, 15]. However, these algorithms regard models as blackboxes and heuristically explain with their own metrics, which cannot give explicit understandings of the actual decision-making process. Different from existing works, we proposed a naturally understandable neural network model.

**Intrinsically Explainable Machine Learning Techniques.** Intrinsically explainable models can be explained without relying on post-hoc algorithms [9, 5], mainly including classic models such as logistic regression [37], linear support vector machine [26], decision trees [43], generalized additive models [25] and Bayesian models [41, 52]. Recently, Agarwal *et al.* [7] proposed Neural Additive Model that predicts with a linear combination of neural networks. However, all these models usually have tight restrictions on the hypothesis space, which limits their fitting ability on complicated real-world problems. Based on these methods, complicated models are developed for higher performance. However, even with intrinsically explainable base models, these complicated models still need post-hoc algorithms for explanation [9]. For example, ensemble tree models [16, 27] predict with a large number of weak learners. However, the joint decision-making process of massive decision trees is difficult to understand. Kernel functions [39] are incorporated in support vector machines to seize high dimensional feature interactions. However, the dimension transformation is implicit and not understandable. In recent years, researchers also try to design explainable neural network models by incorporating purposely designed task-specific constraints or structures [47, 50, 17, 40]. However, these models cannot be adopted by the general tasks. Besides, they only provide heuristic and domain-specific intermediate information instead of telling the complete decision-making process. Different from these works, we aim at a general explainable neural network model, which has an intrinsically explainable decision-making process while retaining the high fitting ability.

## 5 Concluding Remarks

In this paper, we have proposed an explainable deep learning model, VOTEN. Specifically, we theoretically analyzed the major components of VOTEN and discussed its priority over MLP, and accordingly proposed some efficient algorithms to raise the model usability. Experimental results on multiple real-world datasets clearly demonstrated that VOTEN can significantly improve the explainability and performance of deep learning.

**Limitations.** In this paper, we focused on comparing VOTEN with MLP, which is the generic and basic structure of deep learning models. Many powerful problem-specific structures can be derived from MLP by adding operations such as weight sharing (e.g., CNN). Similar to MLP, VOTEN is a basic and generic structure. It can be adopted to problem-specific models (e.g., we can simply use VOTEN to replace MLPs in deepFM [24] or MMoE [36]). Indeed, recent studies show that if properly designed, simple MLP-based structure achieves comparable performance to complicated SOTA models [21, 51]. VOTEN's advantages over MLP provides great possibility on further improving a wide range of deep learning applications. In the future, we will also explore building VOTEN-based task-specific structures. In addition, since VOTEN automatically extracts concepts during training, human effort is needed to observe the voting functions for understanding the concepts, which is a common issue in unsupervised concept modeling, such as Latent Dirichlet Allocation [13]. In the future, we will work on easing the concept understanding of VOTEN, such as recognizing concept-related samples or aligning VOTEN with human-understandable concepts.

## Acknowledgments and Disclosure of Funding

The research work supported by the National Key Research and Development Program of China under Grant No. 2017YFB1002104, the National Natural Science Foundation of China under Grant No. U1836206, U1811461, 62176014, 91746301, 61836013, 61773361.

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
