# Discerning Decision-Making Process of Deep Neural Networks with Hierarchical Voting Transformation

## Appendix

Supplementary Proofs

Supplementary Algorithm: Decision Path Recognition

Supplementary Algorithm: Propagation-based Relevance Analysis

Supplementary Algorithm: VOTEN Pruning

Supplementary Description: Experimental Setup

Supplementary Visualization 1: Global Decision Path

Supplementary Visualization 2: Voting Function

Supplementary Visualization 3: Local Decision Path

Supplementary Visualization 4: Propagation-based Relevance Analysis

Supplementary Visualization 5: Single-sighted Prediction

Supplementary References

35th Conference on Neural Information Processing Systems (NeurIPS 2021).

# A Supplementary Proofs

**Theorem 1** *MLP can be derived from a subset of degenerated VOTEN models whose voting functions in the form of $f_{i,j}^d(x_i^d) = w_{i,j}^d \sigma(x_i^d) + b_{i,j}^d$, where $\sigma$ is a predefined activation function, the scalars $w_{i,j}^d, b_{i,j}^d \in \mathbb{R}$ are trainable parameters.*

*Proof.* Given any $\mathcal{M} \in \text{MLP}$, without loss of generality, we suppose $\mathcal{M}$ has $D$ layers where the feature transformation in each layer formulated as

$$z_j^{d+1} = \sigma(\sum_{i=1}^{n_d} w_{i,j}^d z_i^d + b_j^d), \tag{1}$$

where $\sigma$ denotes the activation function, $\boldsymbol{z}^d = [z_1^d, z_2^d, \cdots, z_{n_d}^d]^{\mathrm{T}}$ denotes the features in the $d$-th layer, $w_{i,j}^d$ and $b_j^d$ are the trainable parameters. In particular, $\boldsymbol{z}^0$ denotes the input feature vector. The model prediction can be written as $F^{MLP}(\boldsymbol{z}^D)$.

We can construct a VOTEN model $\mathcal{M}'$ whose voting functions in the form of $f_{i,j}^d(x_i^d) = w_{i,j}^d \sigma(x_i^d) + b_{i,j}^d$ and has exactly the same modeling process as $\mathcal{M}$, Specifically, $\mathcal{M}'$ has $D$ voting layers. For simplicity, we suppose each voting channel has the same importance, i.e., $a_{i,j}^d = \frac{1}{n_d} \quad \forall d = 0, \cdots, D-1$. In the first layer, the voting function is formulated as

$$f_{i,j}^0(x_i^0) = n_0 w_{i,j}^0 \sigma^I(x_i^0) + b_j^0, \tag{2}$$

where $\boldsymbol{x}^0 = \boldsymbol{z}^0$ is the input feature vector, $\sigma^I$ is the invariant transformation (i.e., $\sigma^I(x) = x$), which we regard as a predefined activation function. Then, for each concept $\mathcal{C}_j^1$, we have

$$x_j^1 = \sum_{i=1}^{n_0} a_{i,j}^d f_{i,j}^0(x_i^0) = \sum_{i=1}^{n_0} \frac{1}{n_0}(n_0 w_{i,j}^0 x_i^0 + b_j^0) = \sum_{i=1}^{n_0} w_{i,j}^0 z_i^0 + b_j^0. \tag{3}$$

According to Equation 1, we have

$$\boldsymbol{z}^1 = \sigma(\boldsymbol{x}^1). \tag{4}$$

In the next $D-1$ layers, we formulate the voting functions as

$$f_{i,j}^d(x_i^d) = n_d w_{i,j}^d \sigma(x_i^d) + b_j^d, \qquad d = 1, \cdots, D-1. \tag{5}$$

Then, the concepts of each layer can be inferred as

$$x_j^{d+1} = \sum_{i=1}^{n_d} a_{i,j}^d f_{i,j}^d(x_i^d) = \sum_i^{n_d} \frac{1}{n_d}(n_d w_{i,j}^d \sigma(x_i^d) + b_j^d) = \sum_i^{n_d} w_{i,j}^d \sigma(x_i^d) + b_j^d. \tag{6}$$

With Equation 6, when we have $\boldsymbol{z}^d = \sigma(\boldsymbol{x}^d)$, we can induce $\boldsymbol{z}^{d+1} = \sigma(\boldsymbol{x}^{d+1})$, because

$$z_i^{d+1} = \sigma(\sum_i^{n_d} w_{i,j}^d z_i^d + b_j^d) = \sigma(x_i^{d+1}).$$

With mathematical induction from Equation 4, we have $\forall d \in 1 \cdots, D$,

$$\boldsymbol{z}^d = \sigma(\boldsymbol{x}^d). \tag{7}$$

Finally, we formulate the final prediction of $\mathcal{M}'$ as

$$F^{VOTEN}(\boldsymbol{x}^D) = F^{MLP}(\sigma(\boldsymbol{x}^D)) = F^{MLP}(\boldsymbol{z}^D), \tag{8}$$

This means for any legal input, the two networks have the same outputs. So far, $\mathcal{M}$ and $\mathcal{M}'$ have exactly the same parameters, intermediate features and final outputs. We can conclude that $\mathcal{M}$ and $\mathcal{M}'$ completely equal to each other. Obviously, $\forall \mathcal{M} \in \text{MLP}$, we can always find an equivalent model belonging to our specified degenerated VOTEN model set, which proves the theorem.

**Definition 1** *In VOTEN, we refer to two concept functions $g$ and $g'$ as equivalent iff. there exists an invertible function $\Phi$, so that $\forall \boldsymbol{x}^0 \in \mathbb{R}^{n_0} \ \Phi(g(\boldsymbol{x}^0)) = g'(\boldsymbol{x}^0)$.*

**Theorem 2** *In VOTEN, if replacing a concept function with an equivalent form, there exists a way to replace its voting functions so that all the downstream concept functions stay unchanged.*

*Proof.* Without loss of generality, we suppose replacing the concept function $g_i^d$ with an equivalent form $g'$. According to Definition 1, we can find an invertible function $\Phi$ so that $\forall x^0 \in \mathbb{R}^{n_0}, \Phi(g_i^d(x^0)) = g'(x^0)$. Since $\Phi$ is invertible, we have $\forall x^0 \in \mathbb{R}^{n_0}, g_i^d(x^0) = \Phi^{-1}(g'(x^0))$. Then, for any concept in the next layer, we have

$$g_j^{d+1}(x^0) = \sum_{i=1}^{n_d} a_{i,j}^d f_{i,j}^d(g_i^d(x^0)) = \sum_{i=1}^{n_d} a_{i,j}^d f_{i,j}^d(\Phi^{-1}(g'(x^0))).$$

Therefore, we can replace the voting function $f_{i,j}^d$ with $f_{i,j}^d \circ \Phi^{-1}$. Then the concept function of $g_j^{d+1}(x^0)$ stays unchanged.

# B Supplementary Algorithm: Decision Path Recognition

## B.1 Decision Path

In VOTEN, the observations are hierarchically transformed into higher-level concepts and finally contribute to the final prediction. An observation can influence the prediction in various patterns. We reveal these patterns with *decision paths*. Formally, we define a decision path as a sequence of connected voting channels that link an input to the final output. Different decision paths may have different importance for the model. By finding out the important decision paths, the model's decision logic can be easily revealed.

## B.2 Influencing Score

We first measure an influencing score $c_{i,j}^d$ for each voting channel $\mathcal{V}_{i,j}^d$. Specifically, we design two kinds of influencing score measurements according to our analysis on the effect of votings, including a local influencing score and a global influencing score. The local influencing score is used to recognize decision paths that are important for specific instances. We calculate the signed local influencing score as $c_{i,j}^d = \frac{a_{i,j}^d f_{i,j}^d (x_i^d) - \overline{h_{i,j}}}{\sum_{k=1}^{n_d} |a_{k,j}^d f_{k,j}^d (x_k^d) - \overline{h_{k,j}}|}$. Here, we conduct normalization to ensure the influencing score to concepts with different value scales are comparable. The sign of the influencing score indicates if the channel positively or negatively contributes to the concept. Then, we calculate the absolute value of the influencing score, i.e., the unsigned influencing score, for filtering important voting channels in our decision path recognition algorithm. The global influencing score is used to recognize generally important decision paths. According to our voting importance analysis, we calculate the variance of the voting channels for importance measurement. Similar with local influencing score, we conduct normalization and estimate the global influencing score as $c_{i,j}^d = \frac{(a_{i,j}^d \sigma_{i,j}^d)^2}{\sum_{k=1}^{n_d} (a_{k,j}^d \sigma_{k,j}^d)^2}$.

## B.3 Recursive Path Filtering

So far, we can measure the importance of voting channels for their target concepts. However, voting channels important for a target concept are not necessarily important for the final output. The reason is that the target concept still votes for the next-layer concepts before it contributes to the prediction. Then, only when the target concept is important for the final output, we can regard a voting channel that is important for it as also important for the final output. We define a concept as important if there exists a path linking it and the output where all the voting channels are important. Then, it can be recursively induced that a concept that votes an important voting channel is important. Therefore, our algorithm finds important decision paths by recursively filter the important concepts and voting channels.

Initially, we regard the output concept of the final layer as important. Recursively, supposing concept $\mathcal{C}_j^{d+1}$ is important, we rank the voting channels to it with their influencing score and find the smallest $K$ so that $\sum_{k=1}^{K} c_{r_k,j}^d \leq \theta_s$, where $\theta_s$ is a pre-defined threshold, $\mathcal{V}_{r_k,j}^d$ denotes the $k$-th important voting channel for $\mathcal{C}_j^{d+1}$. Accordingly, $\mathcal{C}_{r_k}^d$ denotes the $k$-th important voter for $\mathcal{C}_j^{d+1}$. Among the top-$K$ voters, we remove the voters $\mathcal{C}_{r_k}^d$ and their voting channels $\mathcal{V}_{r_k,j}^d$ if their score is smaller than a specific ratio of the maximum score, written as $c_{r_k,j}^d < \theta_r c_{r_1,j}^d$. Then, we regard the rest of the top-$K$ voters and voting channels as important. We repeat this process until we reach the input features. Then, the inputs are connected with the model output through important decision paths, which consist of important concepts and voting channels. In particular, when applying the algorithm, we can gradually increase the thresholds for controlling the granularity of analysis and filter paths from more important to less important. The detailed process is shown in Algorithm 1.

**Algorithm 1** Decision Path Recognition.

**Require:** $\theta_s$: Threshold of the total score. $\theta_r$: Threshold of the minimum score ratio.

**Ensure:** $V$: Set of voting channels forming important decision paths;

1: $V \leftarrow \emptyset$;
2: Add output concepts to an empty set $Q$;
3: **for** $d = D \rightarrow 1$ **do**
4:     **for** each $\mathcal{C}_j^d \in Q$ **do**
5:         $Q' \leftarrow \emptyset$;
6:         **for** $i = 1 \rightarrow n_{d-1}$ **do**
7:             $c_{i,j}^{d-1} \leftarrow$ Influencing score of $\mathcal{V}_{i,j}^{d-1}$;
8:         $\mathcal{R} \leftarrow$ Rank $\{\mathcal{V}_{k,j}^{d-1}\}_{k=1}^{n_{d-1}}$ with influencing score;
9:         $scoreSum \leftarrow 0$;
10:        $scoreMax \leftarrow$ Score of the first element in $\mathcal{R}$;
11:        **for** each $\mathcal{V}_{i,j}^{d-1} \in \mathcal{R}$ **do**
12:            **if** $c_{i,j}^{d-1} < \theta_r * scoreMax$ **then**
13:               Break;
14:            $Q' \leftarrow Q' \cup \{\mathcal{C}_i^{d-1}\}$;
15:            $V \leftarrow V \cup \{\mathcal{V}_{i,j}^{d-1}\}$;
16:            $scoreSum \leftarrow scoreSum + c_{i,j}^{d-1}$;
17:            **if** $scoreSum > \theta_s$ **then**
18:               Break;
19:         $Q \leftarrow Q'$;
    **return** $V$;

## C  Supplementary Algorithm: Propagation-based Relevance Analysis

We can easily find important inputs by observing the filtered decision paths. Furthermore, we design a relevance propagation-based [4] method to quantify the features' relevance to the output.

For each concept $\mathcal{C}_j^d$, we aim to calculate its relevance $R(x_j^d)$ to the output. First, we assign the relevance of the output concept as 1. Then, we back-propagate the relevance to lower-level concepts layer-by-layer. Specifically, each concept's relevance score is allocated to its voters along the voting channels. As we have discussed before, the voting deviation decides the voting effect. Therefore, different from MLP relevance propagation algorithm [4], we design relevance propagation rules for VOTEN based on the voting deviation. Formally, the relevance of $\mathcal{C}_i^{d-1}$ allocated from $\mathcal{C}_j^d$ is written as

$$R(x_j^d)(i) = R(x_j^d) \frac{a_{i,j}^{d-1}(f_{i,j}^{d-1}(x_{i,j}^{d-1}) - \overline{h_{i,j}^{d-1}})}{\sum_{k=1}^{n_{d-1}} a_{k,j}^{d-1}(f_{i,j}^{d-1}(x_{i,j}^{d-1}) - \overline{h_{k,j}^{d-1}})}.$$

For each concept, the relevance is the summation of the relevance score propagated from all the next-layer concepts, written as

$$R(x_i^{d-1}) = \sum_{j=1}^{n_d} R(x_j^d)(i).$$

Since VOTEN has fewer hidden concepts and shorter decision paths, the relevance are allocated and combined much fewer times, which largely reduces error accumulation and guarantees accuracy.

# D  Supplementary Algorithm: VOTEN Pruning

## D.1  Concept estimation with absent voting channels

Intuitively, when some voters are absent for a concept, we have several ways to estimate the concept value only with the rest voters. One way is to get the weighted average among the rest of the voters. That is, we can reallocate the importance of the absent voting channels to the others and formulate the concept estimation as

$$\hat{x}_j^{d+1} = \frac{\sum_{i=1}^{n_d} I_{i,j}^d a_{i,j}^d f_{i,j}^d(x_i^d)}{\sum_{i=1}^{n_d} I_{i,j}^d a_{i,j}^d} = \frac{\sum_{i=1}^{n_d} I_{i,j}^d a_{i,j}^d (f_{i,j}^d(x_i^d) - \overline{h_{i,j}^d})}{\sum_{i=1}^{n_d} I_{i,j}^d a_{i,j}^d} + \frac{\sum_{i=1}^{n_d} I_{i,j}^d a_{i,j}^d \overline{h_{i,j}^d}}{\sum_{i=1}^{n_d} I_{i,j}^d a_{i,j}^d},$$

where $I_{i,j}^d \in \{0,1\}$ indicates if $\mathcal{V}_{i,j}^d$ is not absent. However, this estimation changes the mean value of the concept and causes shift of the original distribution, which may make the downstream voting functions no longer applicable. According to our analysis, only the voting channels' deviation from the average decide the effect. Therefore, we assume the absent channel votes the basic score regardless of the input. Formally, the value of $\mathcal{C}_j^{d+1}$ is estimated as

$$\hat{x}_j^{d+1} = \sum_{i=1}^{n_d} I_{i,j}^d a_{i,j}^d f_{i,j}^d(x_i^d) + \sum_{i=1}^{n_d} (1 - I_{i,j}^d) a_{i,j}^d \overline{h_{i,j}^d} = \sum_{i=1}^{n_d} I_{i,j}^d a_{i,j}^d (f_{i,j}^d(x_i^d) - \overline{h_{i,j}^d}) + b_j^{d+1}.$$

This keeps the mean value unchanged and makes the network stable.

## D.2  Greedy model pruning

A simple idea for pruning the VOTEN model is to adopt the greedy method. Specifically, we can step-by-step drop the voting channel that has the lowest influence on the model performance. Formally, we use $\mathcal{D}_t$ to denote the set of already pruned voting channels at the $t$-th step. In particular, $\mathcal{D}_0 = \emptyset$. Then, we use $P(\mathcal{S})$ to denote the model performance when disabling all the voting channels in a pruned set $\mathcal{S}$. At the $t$-th step, we prune the channel that achieves the highest $P(\{\mathcal{V}_{i,j}^d\} \cup \mathcal{D}_t)$ and update the pruned set as $\mathcal{D}_{t+1} = \{\mathcal{V}_{i,j}^d\} \cup \mathcal{D}_t$. However, for each step, this method needs exhaustively explore the model performance after pruning each voting channel, which is time-consuming.

## D.3  Fast model pruning

We design an efficient approximation algorithm. Instead of pruning the channel that has accurately the least influence at each step, we loosen the constraint and prune the channels which have relatively small influences. Then, we can adopt lazy updating of voting channels' scores during the pruning.

First, we estimate $P(\{\mathcal{V}_{i,j}^d\})$ for each channel as their initial score. Then, we form a sequence of voting channels in score-decreasing order. Intuitively, even though pruning some voting channels may change the score of the rest voting channels, the change may be not always big. The channels originally ranked higher in the sequence are still more likely to have high score. Therefore, we approximate the greedy process. Instead of updating the score of all the voting channels for every time we prune a voting channel, we simply check the channels in the sequence one-by-one and prune a channel if the model performance will not decrease much. To limit performance decrease, we start with a tight performance threshold and gradually loosen it as we prune more and more voting channels. In this way, the model performance will decrease slowly during the pruning process.

Specifically, for each round $t$, we check the sequence in order. Since it is reasonable to assume the score of each channel decrease as the pruning goes, for each voting channel, we check its stored score first. If the score is already lower than the threshold, we can infer that no following channels will satisfy the current constraint. Then, we end the current round and start a new round. If the score is larger than the current threshold, we update its current score to $P(\{\mathcal{V}_{i,j}^d\} \cup \mathcal{D}_t)$. If the updated score is still larger than the current threshold, we prune it and delete it from the sequence. Otherwise, we update its position in the sequence according to its current score.

Moreover, we avoid imbalanced pruning of dropping too many channels for a same concept at a time. For one thing, pruning a voting channel raises the importance of its peers. The score of these voting channels may decrease largely. In this case, skipping these channels can reduce failures of

exploration and raise the algorithm's efficiency. For the other thing, pruning too many channels for the same concept makes the estimation of the concept fragile and more single-sighted. Then, the network may become less robust and more easily corrupt in follow-up pruning. Therefore, for each threshold, we go through the sequence for many rounds. For each round, once we have pruned a voting channel, the other voting channels to the same concept will be skipped.

The detailed algorithm can be found in Algorithm 2.

---

**Algorithm 2** Network pruning for VOTEN.

---

**Require:** $V$: the set of voting channels; $\theta_0$: the initial threshold; $NStep$: the number of times for threshold decrease; $\epsilon$: the step of threshold decrease;

**Ensure:** $\mathcal{S}$: the ordered pruned voting channels;

1: **for** each $\mathcal{V}_{i,j}^d \in V$ **do**
2:      $s_{i,j}^d \leftarrow P(\{\mathcal{V}_{i,j}^d\})$;
3: $\mathcal{R} \leftarrow$ Rank $V$ according to the score.
4: $\theta \leftarrow \theta_0$;
5: **for** $step = 1 \rightarrow NStep$ **do**
6:      $counter \leftarrow 1$;
7:      **while** $counter \geq 1$ **do**
8:          $vis \leftarrow \emptyset$;
9:          $counter \leftarrow 0$;
10:          **for** each $\mathcal{V}_{i,j}^d \in \mathcal{R}$ **do**
11:              **if** $s_{i,j}^d \geq \theta$ and $(d+1, j) \notin vis$ **then**
12:                  $s_{i,j}^d \leftarrow P(\{\mathcal{V}_{i,j}^d\} \cup \mathcal{S})$;
13:                  **if** $s_{i,j}^d \geq \theta$ **then**
14:                      $\mathcal{S} \leftarrow \mathcal{S} \cup \{\mathcal{V}_{i,j}^d\}$;
15:                      $\mathcal{R} \leftarrow \mathcal{R} - \{\mathcal{V}_{i,j}^d\}$;
16:                      $vis \leftarrow vis \cup \{(d+1, j)\}$;
17:                      $counter \leftarrow counter + 1$;
18:                  **else**
19:                      Update the position of $\mathcal{V}_{i,j}^d$ in $\mathcal{R}$;
20:      $\theta \leftarrow \theta - \epsilon$;
     **return** $\mathcal{S}$;

---

# E   Supplementary Description: Experimental Setup

## E.1   Dataset

The experiments are conducted on 4 public anomalous datasets:

**Context-aware Multi-Modal Transportation Recommendation (MR)**[1]. This is the dataset of the KDD CUP competition in 2019, which aims to recommend the most proper transport mode given OD pair and anomalous situational contexts. We extracted features for each record and formulated the recommendation into a multi-classification problem with 11 classes. In particular, we have filtered out the instances that have no clicks. The features mainly contain user portraits and an ordered list of recommended plans of the map app. Each plan consists of transport mode, time, distance, and price. In our experiments, we deleted the first recommended plan's mode information since it is too strongly correlated with the label (many users choose the first recommendation as default). In this way, we can better observe how the model incorporates complicated information for meaningful decisions. The data contains samples collected from Beijing, China, ranged from Oct 1st, 2018 to Nov 30th, 2018. We used samples in the last 7 days for testing and used the rest for training.

**IJCAI-18 Search Conversion Rate Prediction (RP)**[2]. This is the dataset of the IJCAI competition in 2018, which aims to predict if the user will purchase the advertised action given transaction related information. Specifically, this dataset contains massive anomalous transaction records in continuous 7 days collected from taobao.com, a major E-commerce platform in China. Each record contains information about user, advertised item, query, context and shop. We extracted features for each record and formulated the prediction as a binary-classification problem. We used the samples in the last day for testing and used the rest for training.

**Forest Cover Type Prediction (CT) [6]**. This is the dataset for predicting forest cover type from cartographic variables. Each sample is a given observation of forest in a 30 x 30 meter cell and features are obtained from US Geological Survey (USGS) and US Forest Service (USFS) data. We formulated the task as a multi-classification problem with 7 classes. We randomly split the data into training and test set with a ratio of 4:1.

**Census-Income Prediction (CI) [11]**. This is the dataset for predicting if the income will be above 50K, given demographic and employment related features. This dataset contains anomalous census data extracted from the 1994 and 1995 Current Population Surveys conducted by the U.S. Census Bureau. We formulated the task as a binary-classification problem. The original data has been split into training and test set with a ratio of 2:1, which we have followed in our experiments.

**Allstate Claim Prediction (AS) [1].** This is the dataset for predicting Bodily Injury Liability Insurance claim payments based on the characteristics of the insured's vehicle. The data range from 2005 to 2007. We followed [9] and used the last 1,000,000 samples as test set.

**Higgs boson dataset (HG) [5].** This is a classification problem to distinguish between a signal process which produces Higgs bosons and a background process which does not. The first 21 features are kinematic properties measured by the particle detectors in the accelerator. The last seven features are functions of the first 21 features; these are high-level features derived by physicists to help discriminate between the two classes. The last 500,000 examples are used as a test set, which we have followed in our experiments.

The summary of the datasets are listed in Table 1. We have transformed each task into standard prediction tasks with structured input features. Specifically, each sample consists of an input feature vector and a class label.

## E.2   Baselines

The baselines of this work can be grouped into three categories, including:

- Classical tree-based machine learning models, including Decision Tree [12] and Random Forest [13]. These models have been widely-adopted in various data-mining tasks for decades.

---

[1]https://dianshi.bce.baidu.com/competition/29/question
[2]https://tianchi.aliyun.com/competition/entrance/231647/information

Table 1: Summary of the datasets.

| Dataset | Task | Train | Test | Features | Classes |
|---------|------|-------|------|----------|---------|
| MR | Multi-classification | 403,469 | 49,867 | 113 | 11 |
| RP | Binary-classification | 342,432 | 57,418 | 119 | 2 |
| CT | Multi-classification | 464,809 | 116,203 | 54 | 7 |
| CI | Binary-classification | 199,523 | 99,762 | 39 | 2 |
| HG | Binary-classification | 10,000,000 | 500,000 | 28 | 2 |
| AS | Binary-classification | 12,184,290 | 1,000,000 | 31 | 2 |

- Boosting model. We used lightGBM [9], which is the most widely adopted implementation for Gradient Boosting Decision Tree (GBDT) in recent years. It is commonly adopted in top solutions of data-mining contests and soon becomes popular in real-world applications.

- Multi-Layer Perceptron (MLP) [7], which is the basic form of deep learning model. We used MLP as a major comparison and show the advantages of VOTEN over it. We tuned the parameters of MLPs for best performance. **Neural Additive Model(NAM)** [3], which is a newly proposed explainable neural network model. NAM predicts with a linear combination of neural networks.

### E.3   Configuration

For tree-based and boosting models, we referred to generally adopted settings in data-mining contests [2, 15]. For MLP models, we have carefully tuned the super parameters on each dataset to find the suitable depth and number of hidden layers. The experimental results on parameter tuning are shown in Figure 1, where we show the change of performance under different settings of depth and width, respectively. For VOTEN, we tuned to find two sets of parameters for each dataset. The first achieves the highest performance to show that VOTEN is more effective than MLP. The second achieves comparable performance as MLP with the least feature combination, which we use to show VOTEN's advantages on interpretability over MLP. The detailed model structures are shown in Table 2. As two factors (ie., $\alpha$ and $\sigma$) jointly decides the influence of voting channels, to ease the analysis and avoid coupled effect of different factors, we fix $\alpha_{i,k}^d = \frac{1}{n_d}$ for each layer. The weights of the neural network models were randomly initialized with normal initializer [8]. We used LeakyRELU [14] as the activation function. For optimization, we used Adam optimizer [10]. The input features have been normalized in prior. We adopted weight sharing structure for the voting networks to reduce model complexity and ease the experiments. Specifically, voting channels from the same voter is modeled with a unified voting network that has multiple outputs for different target concepts. For simplicity, we have adopted the same structure for all the voting networks. We ran our experiments on a computer with Intel(R) Xeon(R) CPU E5-2680 v4 @ 2.40GHz, RAM of 500G, and 1 GeForce RTX 2080 Ti GPU. Our models were implemented with tensorflow 1.15.

Table 2: Detailed Model Structures.

| Dataset | #Feature | #Output | Model | Voting Layer #Concept | Depth | Voting Network |
|---------|----------|---------|-------|------------------------|-------|----------------|
| MR | 113 | 11 | VOTEN$^-$ | 16 | 1 | [64, 96, 64] |
| | | | VOTEN | 16 | 2 | [64, 96, 64] |
| | | | MLP | 128 | 7 | ✗ |
| RP | 119 | 2 | VOTEN$^-$ | 0 | 0 | ✗ |
| | | | VOTEN | 8 | 2 | [64, 96, 64] |
| | | | MLP | 12 | 3 | ✗ |
| CT | 54 | 8 | VOTEN$^-$ | 32 | 2 | [32, 32, 32] |
| | | | VOTEN | 64 | 2 | [32, 32, 32] |
| | | | MLP | 128 | 7 | ✗ |
| CI | 39 | 2 | VOTEN$^-$ | 0 | 0 | ✗ |
| | | | VOTEN | 4 | 2 | [8, 64, 16] |
| | | | MLP | 32 | 2 | ✗ |
| HG | 28 | 2 | VOTEN$^-$ | 8 | 1 | ✗ |
| | | | VOTEN | 64 | 2 | [64, 96, 64] |
| | | | MLP | 64 | 4 | ✗ |
| AS | 31 | 2 | VOTEN$^-$ | 16 | 1 | ✗ |
| | | | VOTEN | 64 | 2 | [64, 96, 64] |
| | | | MLP | 64 | 4 | ✗ |

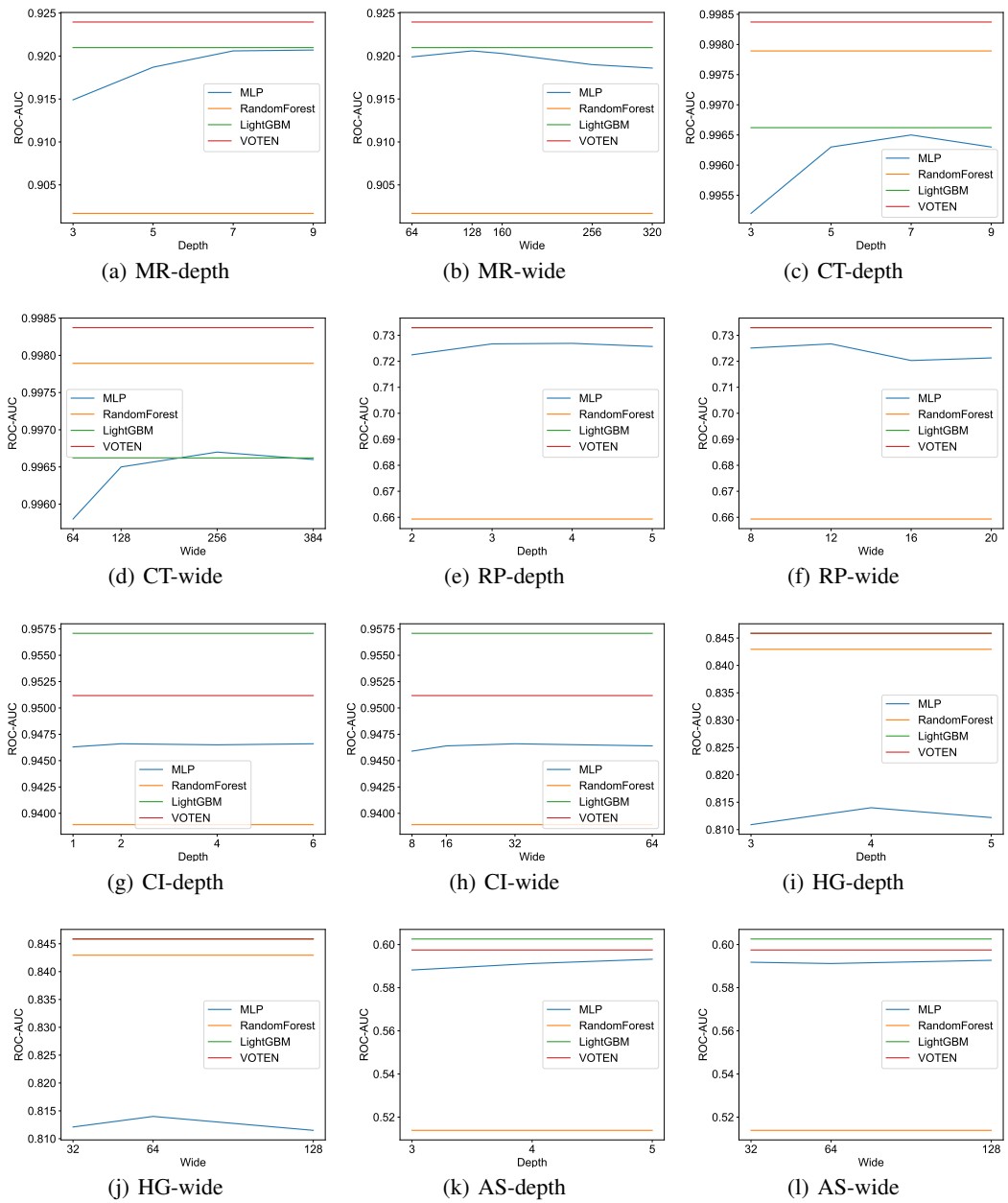

Figure 1: Parameter experiments of MLP.

# F  Supplementary Visualization 1: Global Decision Path

We filter important global decision paths with our proposed decision path recognition algorithm, with $\theta_s = 0.8$ and $\theta_r = 0.1$. The results are shown in Figure 2, Figure 3, Figure 4, and Figure 5. From the figures, we can easily find out important features, concepts, and decision paths that contributes to each class in the global view. These figures also show that VOTEN predicts for different classes with different patterns, in which some concepts may play important roles in multiple patterns. Take MR as an example, L2H7 plays a less important role for class 2 than for class 1. Besides, by comparing the decision paths of calss 1 and class 2, we can find that prediction for class 2 can be influenced by more features than for class 1. Interestingly, although there are 113 features for the MR dataset, the model's decision is mainly based on less than 30 of them. These features are information about the route recommendations by the map app. The results show that the high explainability of MR help distinguishing the less important features. We can easily infer that the model considers the app's route recommendations to be the most important for predicting users' transport mode selection.

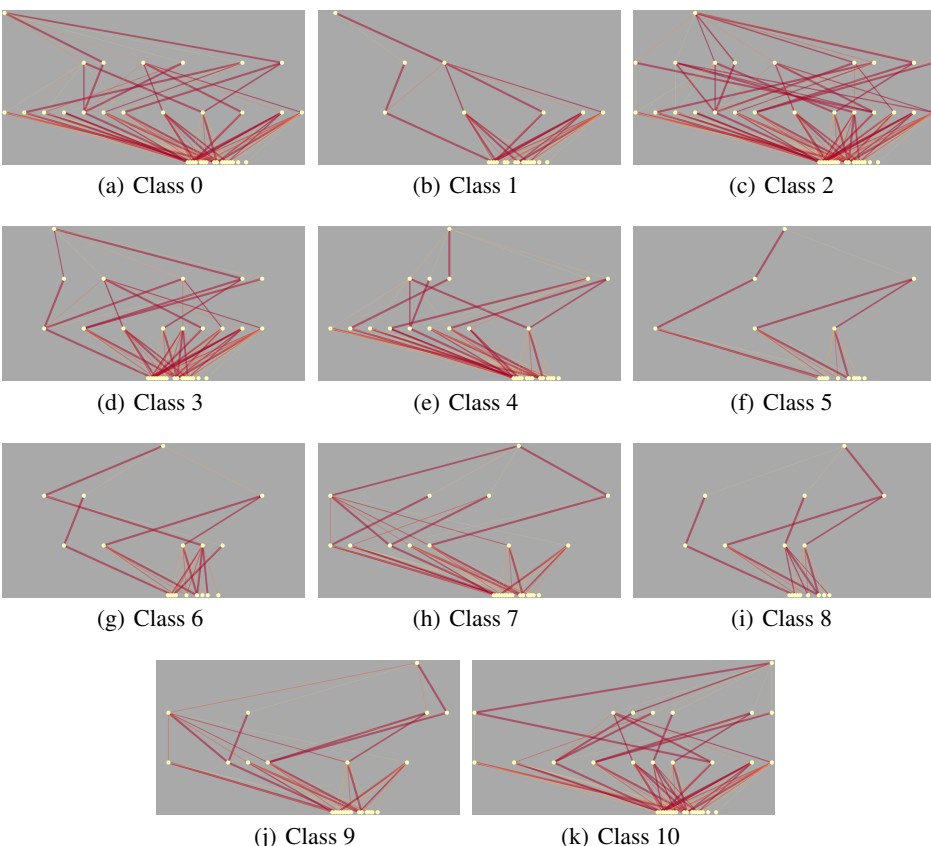

(a) Class 0    (b) Class 1    (c) Class 2

(d) Class 3    (e) Class 4    (f) Class 5

(g) Class 6    (h) Class 7    (i) Class 8

(j) Class 9    (k) Class 10

Figure 2: Global decision path visualization for MR dataset.

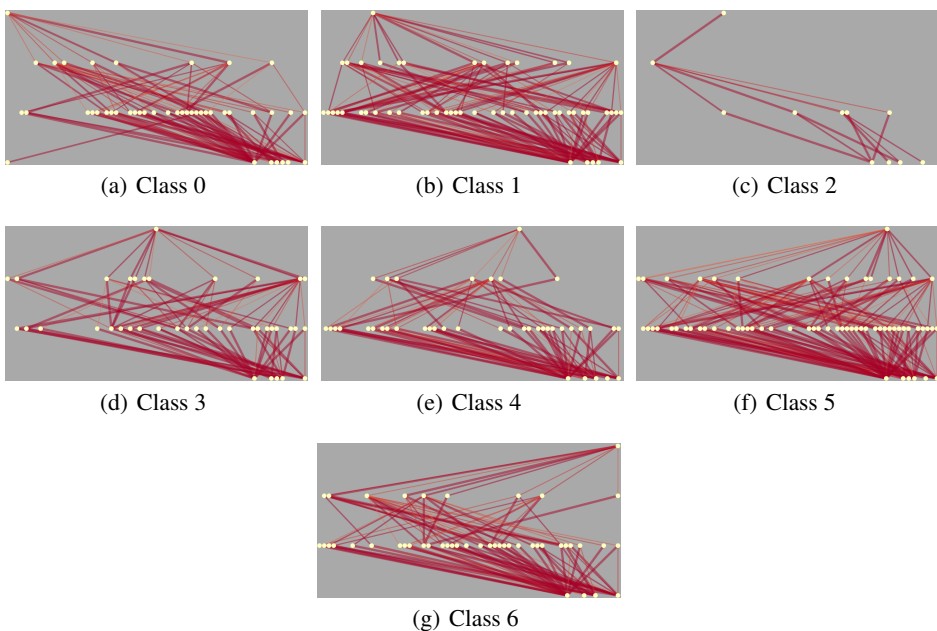

(a) Class 0  (b) Class 1  (c) Class 2

(d) Class 3  (e) Class 4  (f) Class 5

(g) Class 6

Figure 3: Global decision path visualization for CT dataset.

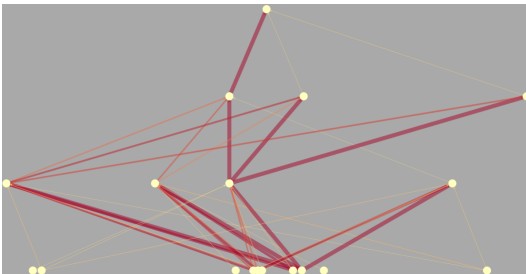

Figure 4: Global decision path visualization for RP dataset.

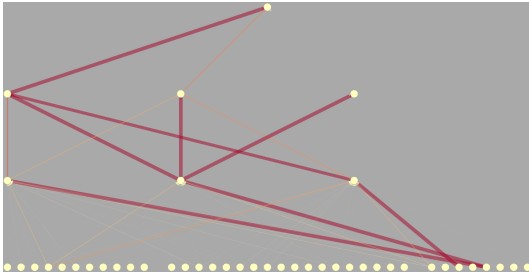

Figure 5: Global decision path visualization for CI dataset.

# G Supplementary Visualization 2: Voting Functions

We visualize the voting functions in VOTEN for different datasets in Figure 6, Figure 7, and Figure 8. Since the voting functions in VOTEN are single-valued, we can easily understand the transformation between concepts. There are several observations. First, voting channels in VOTEN is powerful for modeling complicated nonlinear concept transformations. For example, in Figure 6(i), the voting network models the change of gradient in the transformation. Second, the voting functions are generally smooth and only have a few break points, which makes the voting process explicit and easy to be analyzed. For example, Figure 6(b) is monotonic decreasing and Figure 6(f) has a peak and decreases when the inputs get far from the center. Third, the voting network can distinguish different concepts. For example, according to Figure 6(g) and Figure 6(h), the voting channels from L1H6 to concept L2H9 and L2H11 have totally different functions. Specifically, while L2H9 approximately monotonically decrease with L1H6, L2H11 has a more complicated relationship with L1H6, which has both a valley and a peak in the function.

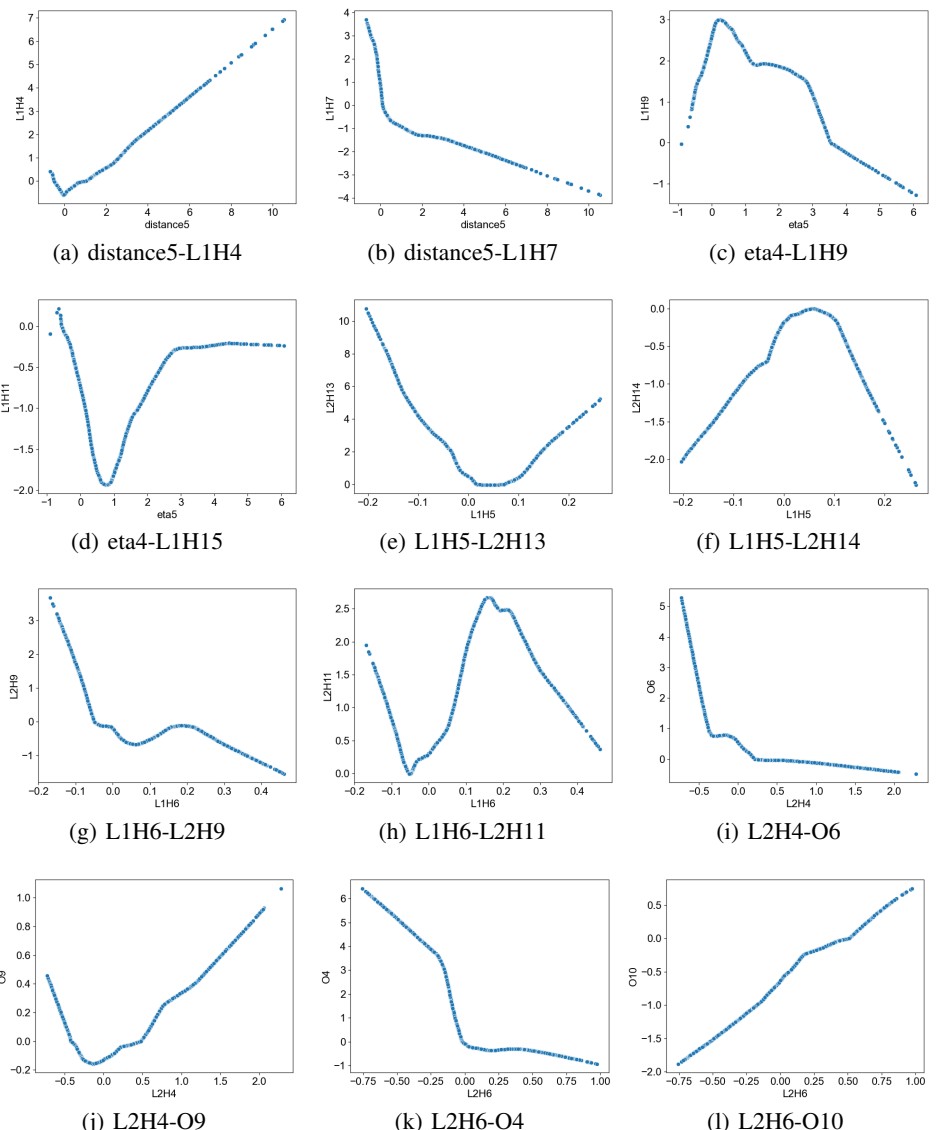

Figure 6: Voting functions in VOTEN trained with MR dataset. x-axis represents concept value and y-axis represents the vote. For the ease of presentation, we represent input features with their names, output node with mark "O", concept $\mathcal{C}_y^x$ in the form of LxHy.

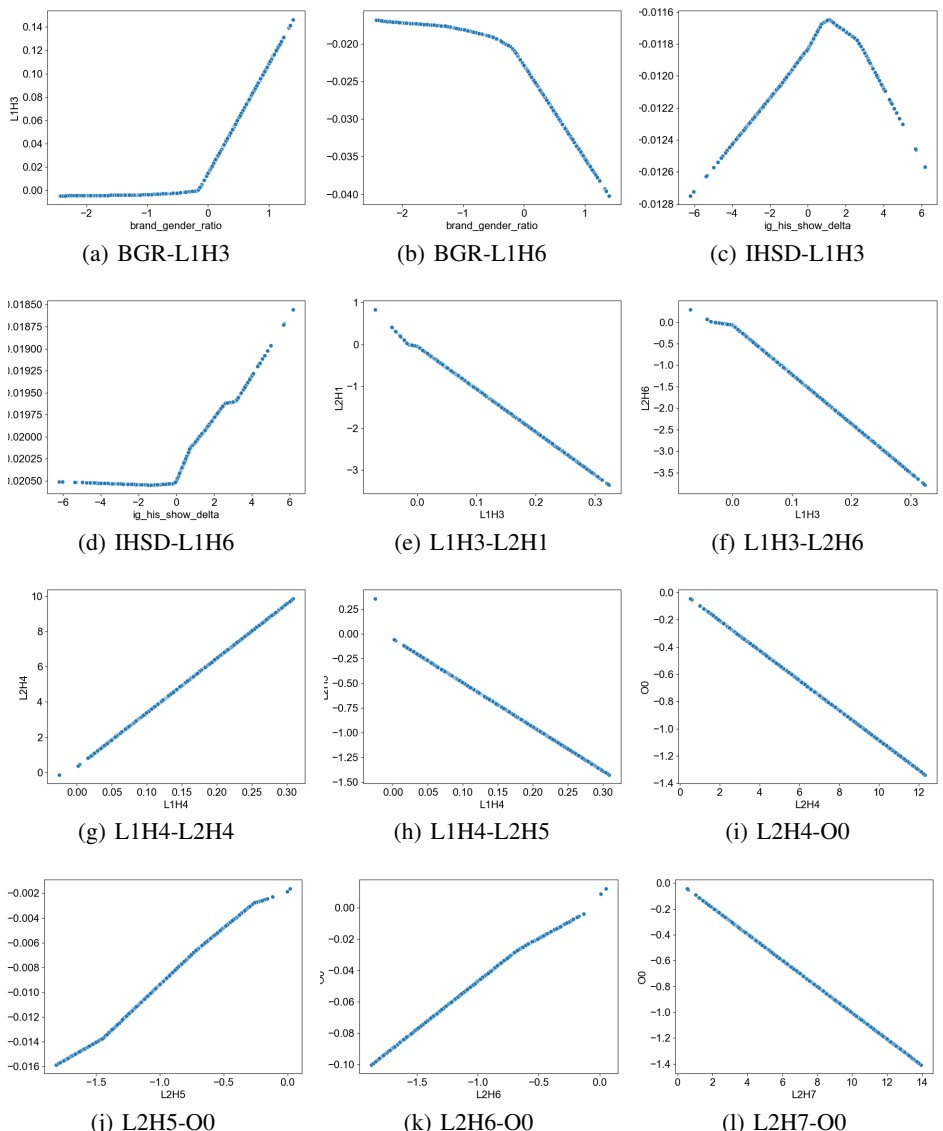

Figure 7: Voting functions in VOTEN trained with RP dataset. x-axis represents concept value and y-axis represents the vote. For the ease of presentation, we represent input features with their names, output node with mark "O", hidden concepts in the form of L$x$H$y$ (the $y$-$th$ concept in the $x$-$th$ voting layer).

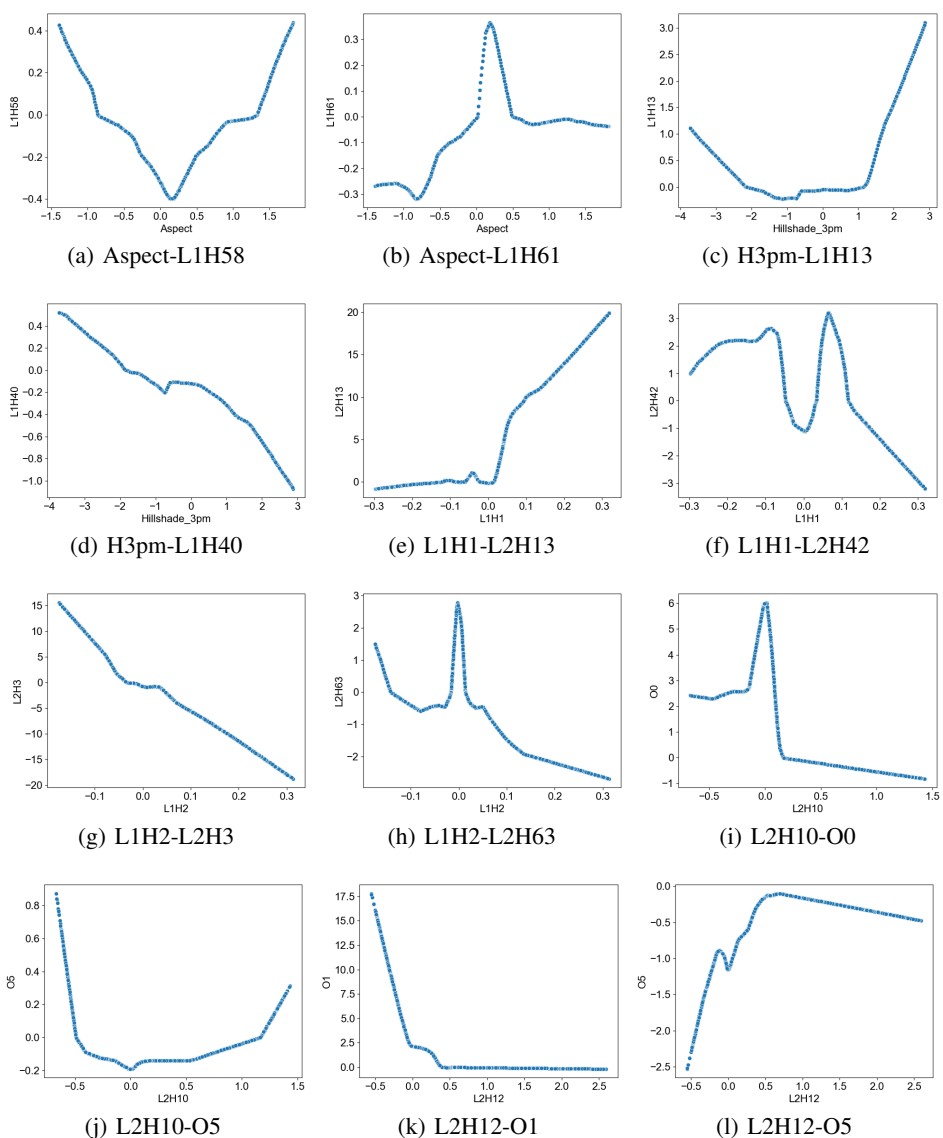

Figure 8: Voting functions in VOTEN trained with CT dataset. x-axis represents concept value and y-axis represents the vote. For the ease of presentation, we represent input features with their names, output node with mark "O", hidden concepts in the form of L$x$H$y$ (the $y$-$th$ concept in the $x$-$th$ voting layer).

# H  Supplementary Visualization 3: Local Decision Path

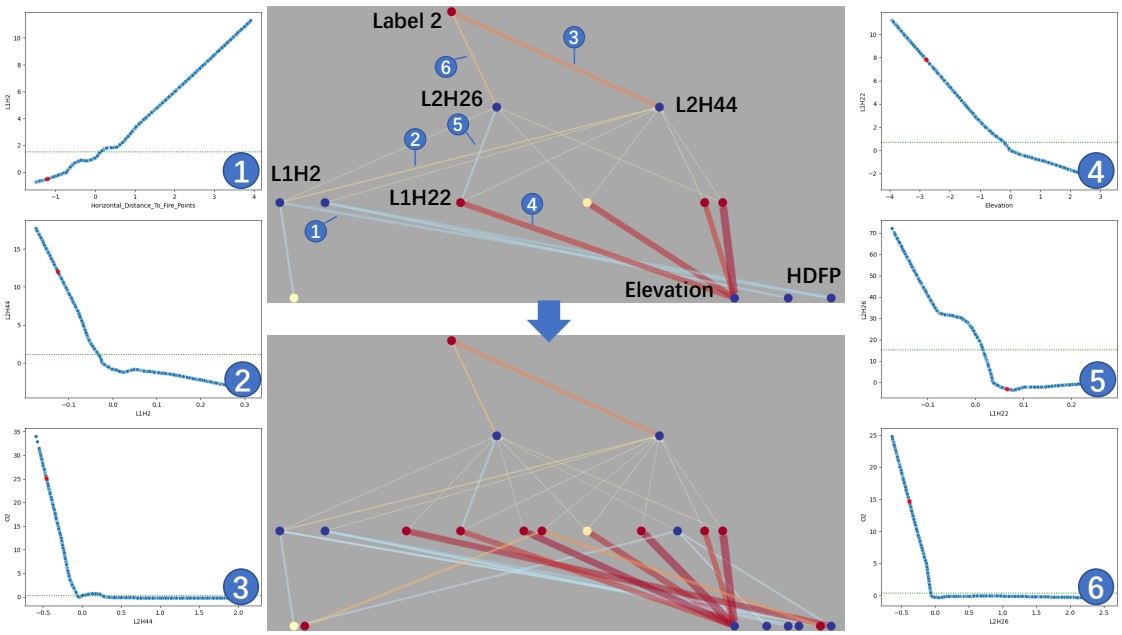

Figure 9: Local decision-making process in CT dataset.

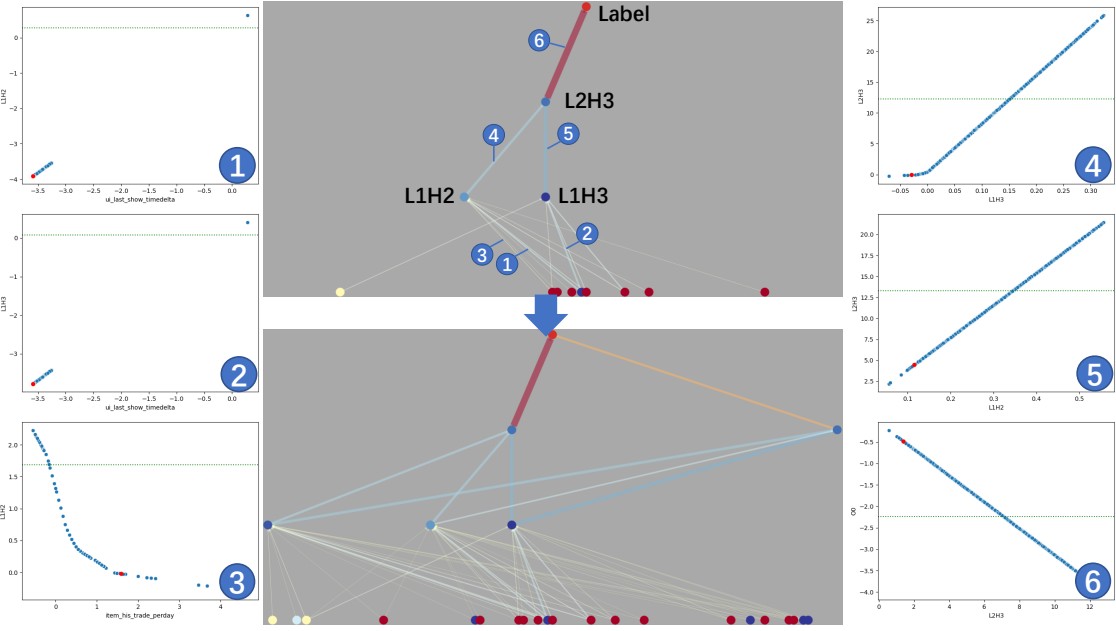

Figure 10: Local decision-making process in RP dataset.

# I Supplementary Visualization 4: Propagation-based Relevance Analysis

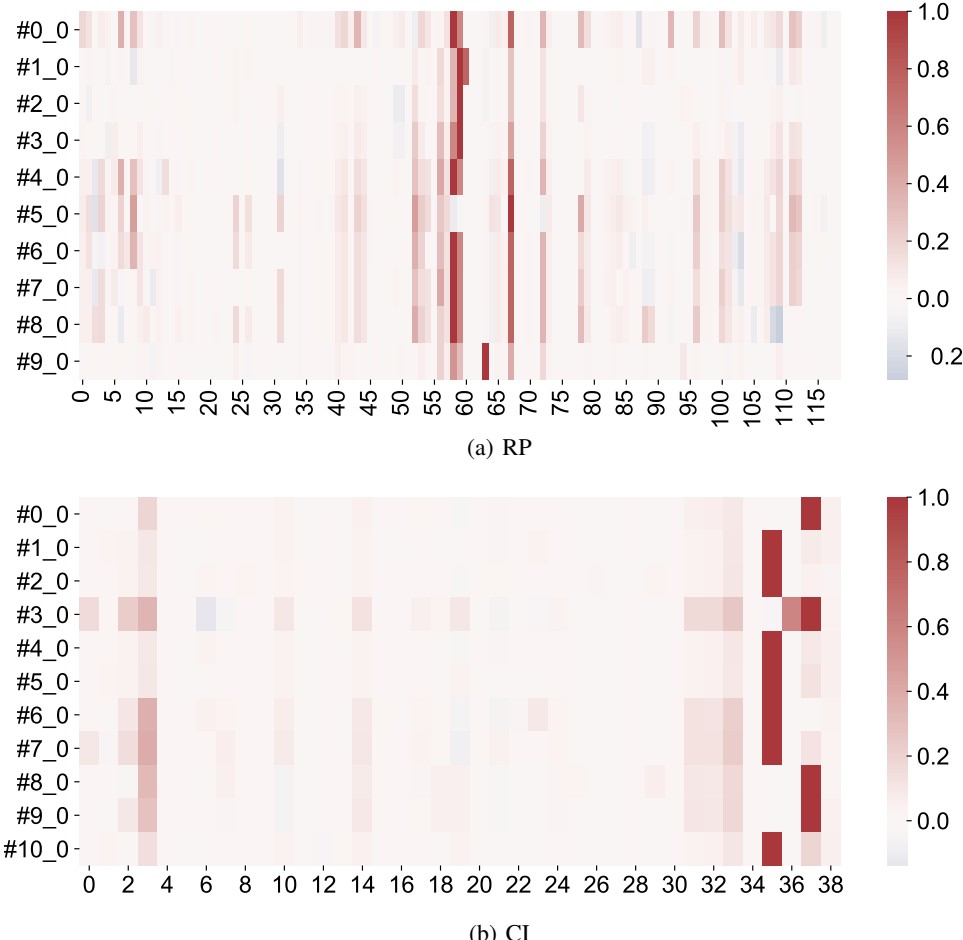

(a) RP

(b) CI

Figure 11: Heatmap for propagation-based relevance in VOTEN trained on different datasets, the x-axis represents features and y-axis represents the instance-output pairs. For ease of presentation, we represent the feature relevance of the $I$-$th$ instance to the $O$-$th$ class as $\#I$-$O$. VOTEN considers more features when making the predictions for PR dataset, which is because most features in PR are statistics strongly correlated with click-through-rates. VOTEN makes full use of these features to make accurate predictions.

# J  Supplementary Visualization 5: Single-sighted Prediction

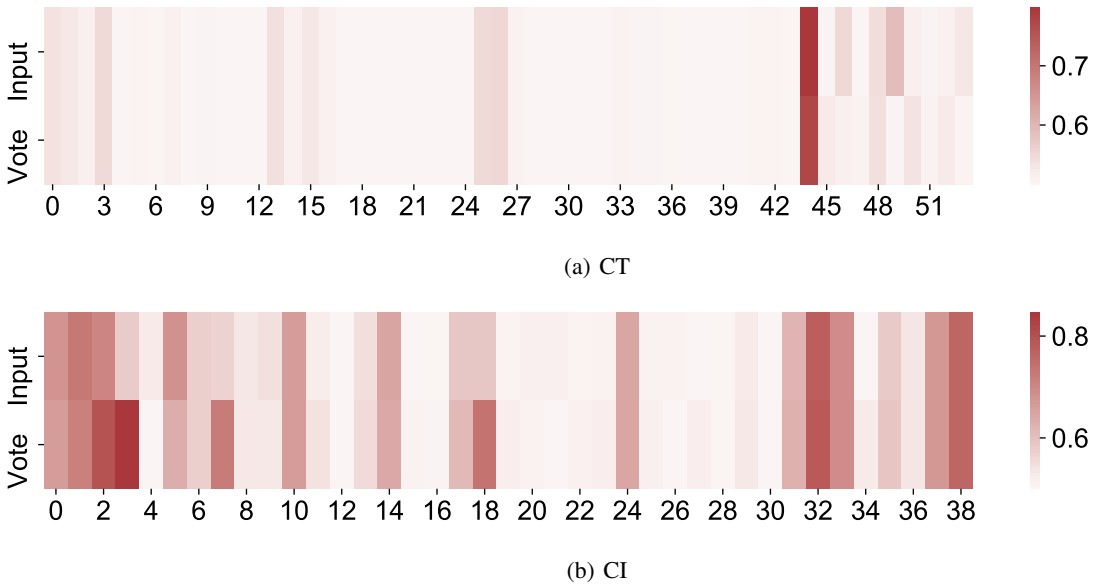

(a) CT

(b) CI

Figure 12: Heatmap for the single-sighted prediction strength in VOTEN trained on different datasets.

We also visualize the estimated functions that transform some single nodes (i.e., features or concepts) to the output score. Indeed, as the single-sighted prediction only involves one input, we can easily observe the predicting function. It can be observed that VOTEN seizes different kinds of influences of features to the output.

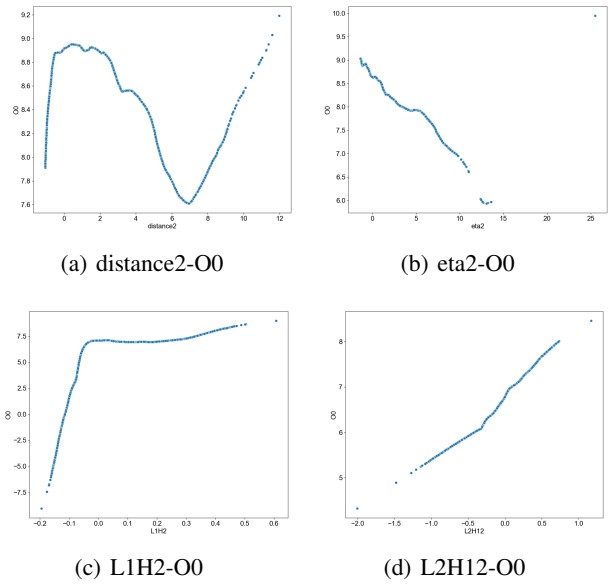

(a) distance2-O0

(b) eta2-O0

(c) L1H2-O0

(d) L2H12-O0

Figure 13: Single-sighted prediction function in VOTEN trained for class 0 in MR dataset.

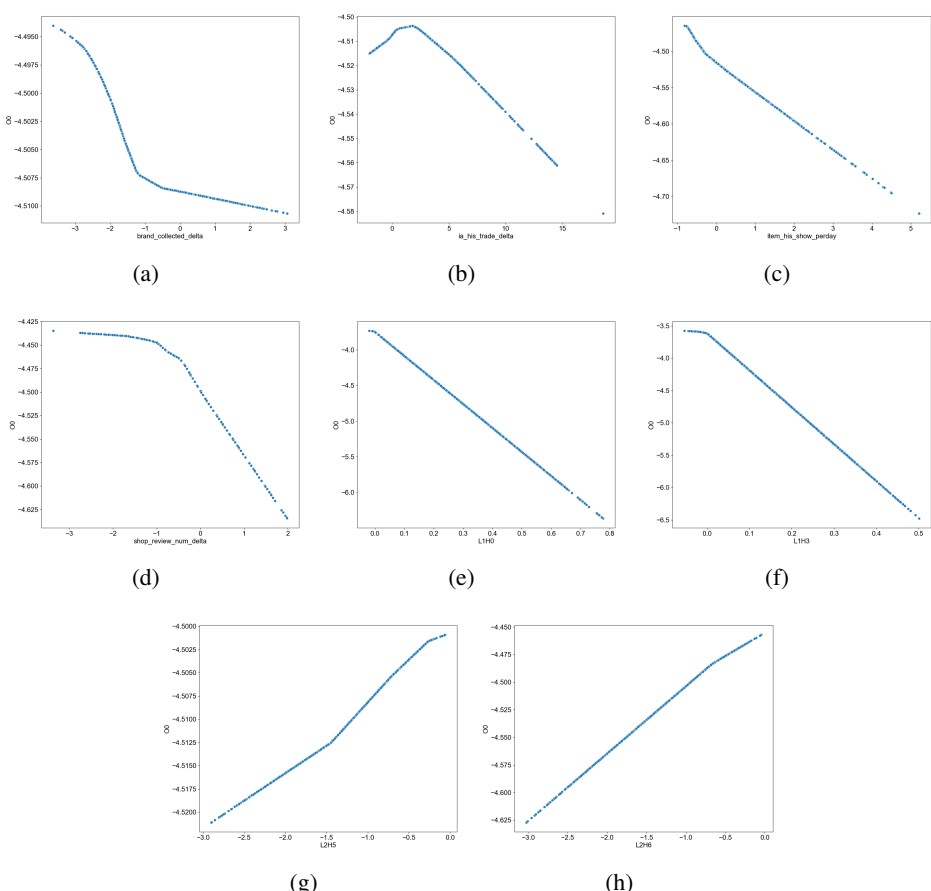

Figure 14: Single-sighted prediction function in VOTEN trained for RP dataset.

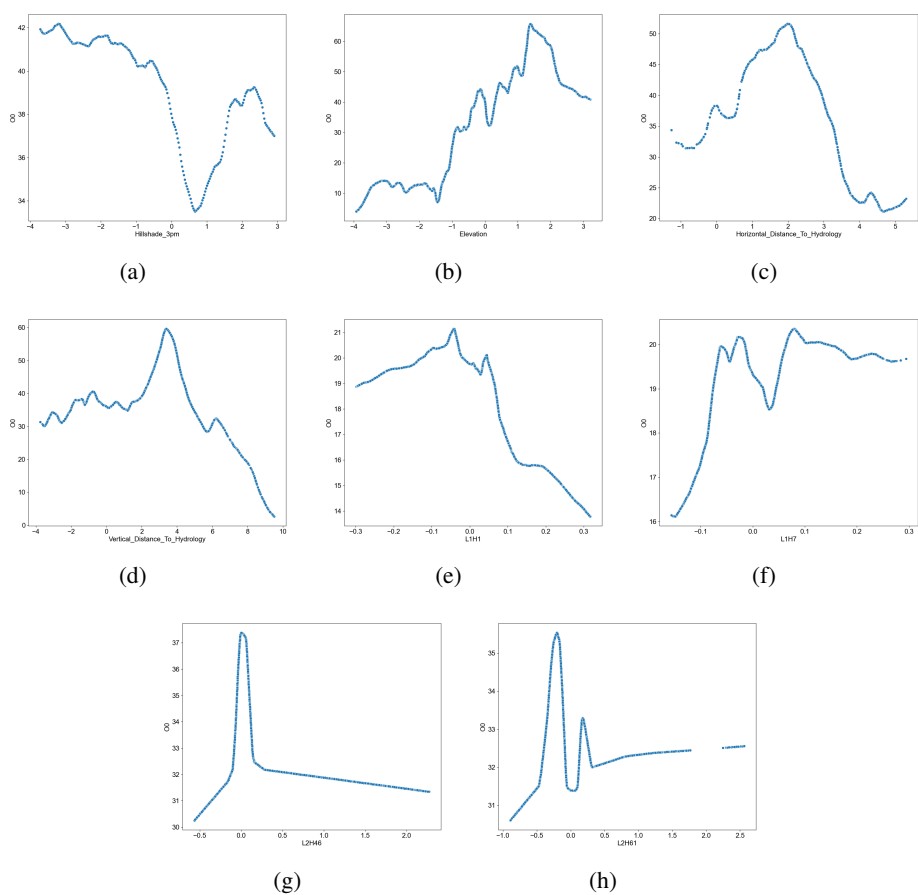

Figure 15: Single-sighted prediction function in VOTEN trained for CT dataset.

## Supplementary References

[1] Allstate claim prediction challenge. `https://www.kaggle.com/c/ClaimPredictionChallenge`.

[2] Ijcai-18 alimama sponsored search conversion rate(cvr) prediction contest. `https://tianchi.aliyun.com/competition/entrance/231647/information`.

[3] Rishabh Agarwal, Nicholas Frosst, Xuezhou Zhang, Rich Caruana, and Geoffrey E Hinton. Neural additive models: Interpretable machine learning with neural nets. *arXiv preprint arXiv:2004.13912*, 2020.

[4] Sebastian Bach, Alexander Binder, Grégoire Montavon, Frederick Klauschen, Klaus-Robert Müller, and Wojciech Samek. On pixel-wise explanations for non-linear classifier decisions by layer-wise relevance propagation. *PloS one*, 10(7):e0130140, 2015.

[5] Pierre Baldi, Peter Sadowski, and Daniel Whiteson. Searching for exotic particles in high-energy physics with deep learning. *Nature communications*, 5(1):1–9, 2014.

[6] Jock A Blackard and Denis J Dean. Comparative accuracies of artificial neural networks and discriminant analysis in predicting forest cover types from cartographic variables. *Computers and electronics in agriculture*, 24(3):131–151, 1999.

[7] Matt W Gardner and SR Dorling. Artificial neural networks (the multilayer perceptron)—a review of applications in the atmospheric sciences. *Atmospheric environment*, 32(14-15):2627–2636, 1998.

[8] Xavier Glorot and Yoshua Bengio. Understanding the difficulty of training deep feedforward neural networks. In *Proceedings of the thirteenth international conference on artificial intelligence and statistics*, pages 249–256, 2010.

[9] Guolin Ke, Qi Meng, Thomas Finley, Taifeng Wang, Wei Chen, Weidong Ma, Qiwei Ye, and Tie-Yan Liu. Lightgbm: A highly efficient gradient boosting decision tree. *Advances in neural information processing systems*, 30:3146–3154, 2017.

[10] Diederik P Kingma and Jimmy Ba. Adam: A method for stochastic optimization. *arXiv preprint arXiv:1412.6980*, 2014.

[11] Nikunj C Oza and Stuart Russell. Experimental comparisons of online and batch versions of bagging and boosting. In *Proceedings of the seventh ACM SIGKDD international conference on Knowledge discovery and data mining*, pages 359–364, 2001.

[12] S Rasoul Safavian and David Landgrebe. A survey of decision tree classifier methodology. *IEEE transactions on systems, man, and cybernetics*, 21(3):660–674, 1991.

[13] Vladimir Svetnik, Andy Liaw, Christopher Tong, J Christopher Culberson, Robert P Sheridan, and Bradley P Feuston. Random forest: a classification and regression tool for compound classification and qsar modeling. *Journal of chemical information and computer sciences*, 43(6):1947–1958, 2003.

[14] Xiaohu Zhang, Yuexian Zou, and Wei Shi. Dilated convolution neural network with leakyrelu for environmental sound classification. In *2017 22nd International Conference on Digital Signal Processing (DSP)*, pages 1–5. IEEE, 2017.

[15] Wenjun Zhou, Taposh Dutta Roy, and Iryna Skrypnyk. The kdd cup 2019 report. *ACM SIGKDD Explorations Newsletter*, 22(1):8–17, 2020.