# OpenReview forum: "Discerning Decision-Making Process of Deep Neural Networks with Hierarchical Voting Transformation"
_NeurIPS.cc/2021/Conference — NeurIPS 2021 Poster_

### Official Review · Reviewer_1htR · 2021-07-04

**Rating:** 8
**Confidence:** 4

**Summary:**

This paper proposes a novel network structure for improving the explainability of deep learning models. The authors theoretically analyzed their model and proved with abundant experiments that the proposed model is effective both in terms of explainability and model performance.

**Limitations And Societal Impact:**

The authors have properly discussed the limitations of their work and discussed the future works for improvement. I look forward to their future works.
I cannot see potential negative societal impact of this work.


**Main Review:**

Generally speaking, this is a well-written paper, and I enjoyed reading it. The proposed model is introduced well in a convincing and insightful way.

The authors give straightforward analysis to show the priority of their model in terms of explainability. Then, they conducted extensive experiments to show the effectiveness of their model. I can see the authors' efforts to improve the usability of their model by providing supportive algorithms and tools. The case studies and visualizations are impressive and show human-understandable inherent classification process of the model. Therefore, I think this work can be a powerful and explainable off-the-shelf solution for classification tasks in many real-world fields.

It is impressive that, while there is usually a trade-off between model explainability and accuracy, the authors have innovatively proposed a new solution to achieve higher performance while assuring high explainability. The authors claimed their model to be generic and can be easily integrated with existing deep learning structures and showed its priority over MLP. Considering more and more MLP-based deep learning models have been achieving SOTA performance in many fields these years, it is exciting to see a powerful substitution with high explainability, which potentially benefits a wide range of deep learning applications.

Still, just like the authors have mentioned in the limitation section, I think the significance of this work can be further enlarged if the authors improve existing task-specific MLP-based models with the proposed structure. I expect the authors in their future work to explore more potentials applications of their model. In addition to models for recommender systems the authors have mentioned (i.e., deepFM and MMoE), models for Computational Vision (e.g., MLP-Mixer) may also be considered.

Besides, the authors have submitted the code for review as supplementary information. Considering its potential influence, I urge the authors to release their code to public platforms as soon as they publish this paper.

The advantage and disadvantage of this paper are summarized as follows:

Advantage:
1. The authors provide a powerful and explainable network structure. The proposed model is generic and can be easily integrated with existing deep learning structures.
2. Impressive experimental results are given to prove the model's effectiveness and explainability.
3. The authors provide a relatively complete workflow for their model's practical use, with carefully designed algorithms and tools for quantifications and visualizations.

Disadvantage:
1. The presentation of this paper needs to be improved. There are typos in the paper, and texts in several figures are small.
2. The significance of this work can be further enlarged if the authors improve existing task-specific MLP-based models with the proposed structure.


**Time Spent Reviewing:**

2

---

> ### Author Response · Authors · 2021-08-10
> **Response to Reviewer 1htR**
>
> ***Comment 1**: Generally speaking, this is a well-written paper, and I enjoyed reading it. The proposed model is introduced well in a convincing and insightful way.*
>
> **Response**: Thank you for your acknowledgment and all your encouraging words.
>
> ***Comment 2**: The authors give straightforward analysis to show the priority of their model in terms of explainability. Then, they conducted extensive experiments to show the effectiveness of their model. I can see the authors' efforts to improve the usability of their model by providing supportive algorithms and tools. The case studies and visualizations are impressive and show human-understandable inherent classification process of the model. Therefore, I think this work can be a powerful and explainable off-the-shelf solution for classification tasks in many real-world fields.*
>
> **Response**: We very much thank you for your recognition on the idea of this paper. To prove the explainability of the VOTEN decision-making process and its usefulness, we did make a large effort conducting these experiments.
>
> ***Comment 3**: It is impressive that, while there is usually a trade-off between model explainability and accuracy, the authors have innovatively proposed a new solution to achieve higher performance while assuring high explainability. The authors claimed their model to be generic and can be easily integrated with existing deep learning structures and showed its priority over MLP. Considering more and more MLP-based deep learning models have been achieving SOTA performance in many fields these years, it is exciting to see a powerful substitution with high explainability, which potentially benefits a wide range of deep learning applications.*
>
> **Response**: Thank you for your acknowledgment of VOTEN’s performance. VOTEN’s structure makes it have a high ability on reasoning from heterogeneous features. Indeed, we have tested VOTEN on several real-world tasks. It turns out that VOTEN achieved far better performance than MLP-based solutions.
>
> ***Comment 4**: Still, just like the authors have mentioned in the limitation section, I think the significance of this work can be further enlarged if the authors improve existing task-specific MLP-based models with the proposed structure. I expect the authors in their future work to explore more potentials applications of their model. In addition to models for recommender systems the authors have mentioned (i.e., deepFM and MMoE), models for Computational Vision (e.g., MLP-Mixer) may also be considered.*
>
> **Response**: Thank you for your constructive comments! Indeed, VOTEN has huge potential for solving various problems. We promise to explore more potential applications with it. It is really an inspiring idea to explore how to adopt VOTEN on homogenous inputs, which no doubt will be an important direction of our future work.
>
> ***Comment 5**: Besides, the authors have submitted the code for review as supplementary information. Considering its potential influence, I urge the authors to release their code to public platforms as soon as they publish this paper.*
>
> **Response**: Thank you for your suggestion! We promise to release all the code if this paper can get accepted.
>
> ***Comment 6**: The presentation of this paper needs to be improved. There are typos in the paper, and texts in several figures are small.*
>
> **Response**: We are sorry for the typos and the small fonts in the figures. We will double-check and correct them in the revised version.

---

### Official Review · Reviewer_4PGX · 2021-07-15

**Rating:** 5
**Confidence:** 4

**Summary:**

This paper proposes a novel type of model, Voting Transformation-based Explainable Neural Network (VOTEN), which represents the transformation from the input to the output to be a hierarchical voting process. This simple structure makes the model interpretable for a human. The papers systematically analyses the advantages of VOTEN, in particular the model's explainability (global and local) and reduced complexity (pruning).

**Main Review:**

I am very much at odds with myself when it comes to this paper. On the one hand it aims to solve a very relevant problem. Through it specific structure VOTEN is relatively easy to interpret while obtaining quite good accuracies, significantly better than e.g. simple decision trees. The experiments presented in the paper are very nice and cover different aspects, e.g., performance, local and global explainability and complexity.
On the other hand I am not fully convinced about the proposed model. First, I find all the discussions claiming that VOTEN is very similar to human reasoning very strange and not appropriate. The authors do not present any results, which support this claim. Furthermore, in the experimental section VOTEN is only applied to relatively simple problems. The big strength of deep neural networks (this to some degree also hold for models such as random forests, SVMs etc.) is that they also perform well on complex, high-dimensional problem. I am not sure that this also holds for VOTEN. The true practical abilities of VOTEN are hard to assess, also because the selected datasets are not the typical benchmark datasets for e.g. GMBs or RandomForest classifiers. Furthermore, there is no comparison with post-hoc explanation methods such as LRP or SHAP applied to a deep neural network or e.g. the work of Lundberg computing Shapley values on tree-based models. Thus, in practice it is unclear how much insights the explanations provided by VOTEN really bring. Also there is no comparison or extensive discussion of the proposed model to the work of Agarwal et al. 2020. Thus, the advantages VOTEN over sota remain unclear.
Overall, because of these limitations I am not fully convinced about VOTEN and therefore see the paper lying below the acceptance threshold.

**Time Spent Reviewing:**

3

---

> ### Author Response · Authors · 2021-08-10
> **Response to Reviewer 4PGX**
>
> ***Comment 1**: I am very much at odds with myself when it comes to this paper. On the one hand it aims to solve a very relevant problem. Through it specific structure VOTEN is relatively easy to interpret while obtaining quite good accuracies, significantly better than e.g. simple decision trees. The experiments presented in the paper are very nice and cover different aspects, e.g., performance, local and global explainability and complexity.*
>
> **Response**: Thanks for your acknowledgment about the broad coverage of our experiments! Your suggestions are constructive and helpful. We have conducted more experiments (which can be found in the following responses) and will add them to our paper. In the following, we would like to respond to your comments point-to-point.
>
> ***Comment 2**: On the other hand I am not fully convinced about the proposed model. First, I find all the discussions claiming that VOTEN is very similar to human reasoning very strange and not appropriate. The authors do not present any results, which support this claim.*
>
> **Response**: As you pointed out, it is inappropriate to claim VOTEN is similar to human reasoning. We will correct this in the revised version. In the paper, we meant to state that VOTEN’s decision-making process can be intuitively understood in a human reasoning way. Please allow us to explain our major idea again:
>
> When predicting with a set of informative heterogeneous features, the models need to quantify the factors’ relationship with the outcome. The way they do it decides if we can understand their decision-making process. Specifically, MLP generates massive coupled intermediate variables with linear transformation layer-by-layer. In contrast, VOTEN generates a few intermediate variables with nonlinear transformation. Indeed, we can understand nonlinear relationships between a small number of variables. For example, we understand that too heavy or skinny may be both unhealthy while the middle part is just good. From this point of view, we say VOTEN (less concept, nonlinear transformation) is more understandable in a human reasoning way than MLP (more concept, linear transformation).
>
> We wanted to show explainability with the case study, which is an example of our generic model understanding process: (1) We analyze the meaning of concepts bottom-to-up by observing the voting functions. Then, we generate a dictionary. It gives each concept an easy-to-understand name and corresponding explanations (like some personality assessments do). (2) Given a predicted sample, we explain up-to-bottom the value of each concept and how they are voted. We can achieve these steps by directly observing VOTEN’s intermediate voting process. Therefore, we say we can naturally understand VOTEN’s decision process in a human reasoning way.
>
> ***Comment 3**: Furthermore, in the experimental section VOTEN is only applied to relatively simple problems. The big strength of deep neural networks (this to some degree also hold for models such as random forests, SVMs etc.) is that they also perform well on complex, high-dimensional problem. I am not sure that this also holds for VOTEN.*
>
> **Response**: Thank you for your comment! There is no doubt that neural networks have developed big advantages in handling high-dimensional problems. On the contrary, they may not be very good at handling these “simple problems”. We actually focused on problems with heterogeneous inputs, where each dimension is an individual feature that has explicit meaning. Compared with images or time series, these kinds of inputs are relatively low-dimensional. However, they are not simple because each individual feature will have different influences on the outcome. The model needs to better model features’ individual effects and figure out how these heterogeneous features are linked with each other. This kind of problem is common in the real world. We often have the need to predict something with a lot of intuitively relevant factors. Many data-mining competitions have been asking the attendees to solve these problems. It turns out the winner solutions are mostly based on tree-based boosting models. This shows neural networks cannot fully utilize the information in these heterogenous features.
>
> Indeed, we came up VOTEN when trying to solve a domain-specific task in an open scenario. Abundant heterogenous features were collected from multiple data sources. We tried many models and found tree-based boosting models are always better. However, we still wanted an NN-based solution for its potential flexibility for complicated operations. For example, we can use domain adaption or multi-task framework to achieve knowledge transfer. We can also combine the model with multi-model inputs (e.g., adding a module for high-dimensional image features). Despite all these advantages, we cannot sacrifice the effectiveness of modeling these heterogeneous features, which are usually the core of the problem. These features are generated from domain knowledge and are usually strongly related to the outcome. More importantly, they are human-understandable factors and can be convincing when explaining the prediction to users.
>
> Therefore, we built VOTEN to utilize these informative features in an effective and explainable way. We did not mean VOTEN to be a substitution for previous neural network structures. But it still makes important progress on handling this kind of common and important problems. In practice, we can combine different network structures to utilize information from various inputs. For example, we can use CNN to extract image features, use VOTEN to reason from heterogeneous features, and then combine them to get more accurate results.
> Thank you for your inspiring suggestion. Considering the recent progress on applying MLP structures on images, we will also explore VOTEN-based neural networks to solve high-dimensional tasks in our future works.
>
> |     | LGB|MLP	|NAM|VOTEN|
> |  ----  | ----  | ----  | ----  | ----  |
> |MR|0.481, 0.921|0.487, 0.920|0.433, 0.898|**0.500**, **0.924**|
> |RP|0.053, 0.731|0.052, 0.725|0.052, 0.729|**0.055**, **0.732**|
> |CT|0.975, 0.997|0.966, 0.997|0.700, 0.950|**0.978**, **0.998**|
> |CI|**0.697**, **0.957**|0.622, 0.945|0.656, 0.951|0.652, 0.951|
> |Allstate|**0.012**, **0.609**|0.011, 0.590|0.011, 0.574|**0.012**, 0.603|
> |HIGGS|0.859, 0.846|0.830, 0.815|0.790, 0.775|**0.862**, **0.848**|
>
>
> ***Comment 4**: The true practical abilities of VOTEN are hard to assess, also because the selected datasets are not the typical benchmark datasets for e.g. GMBs or RandomForest classifiers.*
>
> **Response**: Thank you for your helpful suggestions! Since these baselines used different datasets, we just randomly picked some available data-mining competition datasets. We very much appreciate your suggestion on using benchmark datasets. Following your suggestion, we compared the model performance with “Allstate” and “HIGGS” datasets used by LightGBM. The results are shown in the following table (each entry is shown as AP, AUC). We can observe that VOTEN is comparable with LightGBM on “Allstate” and performs better than LightGBM on “HIGGS”. On both datasets, MLP performs significantly worse than VOTEN. These results are consistent with our observations in the paper.
>
> ***Comment 5**: Furthermore, there is no comparison with post-hoc explanation methods such as LRP or SHAP applied to a deep neural network or e.g. the work of Lundberg computing Shapley values on tree-based models. Thus, in practice it is unclear how much insights the explanations provided by VOTEN really bring.*
>
> **Response**: Thank you for your suggestion and sorry for the confusion! We did not regard previous post-hoc methods as comparisons because we focus on model's self-explainability. VOTEN’s advantage is to enable a direct understanding of the decision-making process. If needed, previous post-hoc algorithms can also be applied to VOTEN to ease the model explanation steps.
>
> ***Comment 6**: Also there is no comparison or extensive discussion of the proposed model to the work of Agarwal et al. 2020. Thus, the advantages VOTEN over sota remain unclear. Overall, because of these limitations I am not fully convinced about VOTEN and therefore see the paper lying below the acceptance threshold.*
>
> **Response**: Thank you for your suggestion! We indeed have tested NAM (Agarwal et al. 2020) in our preliminary experiments, where we have followed the recommended configurations in their paper. We did not use it as a comparison because it did not show competitive performance. Indeed, NAM can also be considered under VOTEN architecture. Specifically, a VOTEN model with no hidden concepts and weights each feature equally will have the same expressiveness as NAM. We can infer that it is difficult for NAM to handle complicated tasks requiring relatively deep feature interactions. We appreciate your suggestion that showing the performances of recent models will make the results more convincing. In the revised version of our paper, we will add these results. NAM’s performance (together with the extra two datasets) can also be found in the table. We can observe that VOTEN performs better since it can model essential feature interactions.

---

### Official Review · Reviewer_v1Ts · 2021-07-18

**Rating:** 3
**Confidence:** 4

**Summary:**

The authors propose an explainable deep learning model that uses a hierarchical voting strategy to perform hierarchical feature selection.

**Limitations And Societal Impact:**

Ye

**Main Review:**

raises predictive performance, decreases feature combinations, supports efficient pruning and feature analysis.

"humans usually aggregate information to infer intermediate concepts step-by-step." This should have a citation.  Really a lot of the information in the first paragraph of 2.1 needs justification through citation.

"That is, while deep neural networks make the decision according to a unified, inseparable complicated function, humans usually aggregate information to infer intermediate concepts step-by-step"  Is this not what many convolutional networks have been shown to do in recent work on interpretation and explainability?

How does VOTEN constrain the model to capture indidual concepts in each module?

In equation 1, why replace C with x?  Is C_d+1 a weighted sum of the output of f(C_d)?  It is not clear at this point in the paper what the function "f" does.  By theorem 1 this is the key difference from a regular MLP so this should be defined sooner.

In general I am not sure what function "f" is in equation 1 or how voting occurs.  Is voting done by outputing a weight for each concept that are then summed to form a new concept?

The approach described in section 3.3 is not very clear and seems difficult to generalize.  Can the authors provide a clear high level description of this process as it would generalize to other settings?

I am not convinced at using correlating decision paths and model activations for explainability as there are many variables to consider.  For the subway example, are thos paths possibly attributed to some other co-ocurring variable with subway in the data?

A related work section should be added to address how this approach relates to other work in network explainability.

**Time Spent Reviewing:**

1.5

---

> ### Author Response · Authors · 2021-08-10
> **Response to Reviewer v1Ts**
>
> Please read our response. We believe your concerns can be addressed as described in the response. Your reconsideration will be greatly appreciated.
>
> ***Comment 1**: "humans usually aggregate … Really a lot of the information in the first paragraph of 2.1 needs justification through citation.*
>
> **Response**: Many thanks for your valuable suggestions! According to your suggestion, in the revised version, we would like to add extra references of literatures for corresponding descriptions.
>
> ***Comment 2**: "That is, while deep neural networks make the decision according to a unified, inseparable complicated function, humans usually aggregate information to infer intermediate concepts step-by-step" Is this not what many convolutional networks have been shown to do in recent work on interpretation and explainability?*
>
> **Response**: In the paper, we tried to introduce the motivation of our model from the view of human understanding. While many neural networks are aggregating information, they are still black boxes for us. It is true that CNN is special for showing human-understandable concepts. As its inputs are pixels, forming concepts with linear aggregation makes sense in our understanding. Also, we can know what feature maps mean by visualizing important pixels as images.
>
> However, for a broader range of tasks whose inputs are heterogeneous features (e.g., size, length, and color of a mushroom), we usually cannot understand the internal information aggregation of neural networks. Specifically, the massive linear combinations of non-linearly-related features bring massive coupled hidden units. It is more difficult to understand what they mean because (1) they cannot be intuitively visualized, and (2) the same set of features can infer different concepts with different transformations. Then, it is inexplicit how intermediate concepts are aggregated in these models. Therefore, we proposed VOTEN to model heterogeneous features, whose intermediate information aggregation is explicit and easy to understand.
>
> We apologize for causing confusion by our oversimplification. In the revised version, we will modify the descriptions to make this clear.
>
> ***Comment 3**: How does VOTEN constrain the model to capture individual concepts in each module?*
>
> **Response**: Sorry for the confusion. Actually, VOTEN can learn to aggregate information into concepts automatically instead of using extra constraints.
>
> ***Comment 4**: In equation 1, why replace C with x? Is C_d+1 a weighted sum of the output of f(C_d)? It is not clear at this point in the paper what the function "f" does. By theorem 1 this is the key difference from a regular MLP so this should be defined sooner.*
>
> **Response**: We are sorry for the confusion. Here C denotes a concept (e.g., “length”) while x denotes the value (e.g., “11cm”) of it.
>
> ***Comment 5**: In general I am not sure what function "f" is in equation 1 or how voting occurs. Is voting done by outputing a weight for each concept that are then summed to form a new concept?*
>
> **Response**: f is a nonlinear single-valued function, modeled with a network with a single input. Its output is the input concept's vote for the value of the estimated concept. When getting the final estimation, the counting layer gets a weighted average of votes from all the voters (lower-level concepts).
>
> ***Comment 6**: The approach described in section 3.3 is not very clear and seems difficult to generalize. Can the authors provide a clear high level description of this process as it would generalize to other settings?*
>
> **Response**: This process can be easily generalized. Generally, the network explanation process we have shown has four steps, including two steps of global explanation and two steps of local explanation. First, we recognize important decision paths globally and find important concepts and their key voters. Second, we analyze the meaning of concepts bottom-to-up by observing the voting functions on the decision paths. Then, we generate a dictionary that gives each concept an easy-to-understand name and corresponding explanations (like some personality assessments do). With the first two steps, we know the model’s whole decision process. Third, we recognize important local decision paths for samples to be analyzed. Fourth, we explain up-to-bottom why the decisions are made.
>
> ***Comment 7**: I am not convinced at using correlating decision paths and model activations for explainability as there are many variables to consider. For the subway example, are those paths possibly attributed to some other co-ocurring variable with subway in the data?*
>
> **Response**: Sorry for the confusion. Actually, we do not use decision paths as a post-hoc explanation method for VOTEN. Instead, VOTEN’s decision-making process is self-explainable. Specifically, as we have introduced, VOTEN is a hierarchical voting process. A sample is recognized to be positive because the voters vote it to be positive. The voters vote like that because their concepts are estimated to be at some specific value range. This value is recurrently voted by the lower-level concepts. This process is exactly how the model makes the prediction. Our model explanation is to objectively display this process by showing the voting functions for each concept. Decision paths do not decide explainability. It is simply an auxiliary tool guiding users to focus on more important voting channels and concepts for easing human analysis. In our system, the thresholds for discovering decision paths can be adjusted so that we can understand the process from general to specific.
>
> Nevertheless, decision path is still an effective tool to filter important voting channels and concepts. If we have understood your concern correctly, you mean variables may have coupling effects on the prediction. Indeed, even two factors are irrelevant to the outcome, if they always co-occur in the dataset, the network may converge to a point where they each influence the prediction (e.g., one positively while the other negatively). Since they co-occur, the prediction will still be correct. In this case, decision path from both factors will be recognized. However, this problem lies in that the network learned the wrong decision logic. The model will make wrong predictions for samples (may be out of the training set) where the two factors no longer co-occur.
>
> Indeed, VOTEN has advantages in handling this kind of noise. First, as we have discussed in section 2.2, VOTEN alleviates the massive intermediate coupling effect of NN. Second, VOTEN makes it difficult for coupling variables to cancel each other out. Specifically, the features go through independent voting networks that are initialized differently. Since the transformations are complex, it is more difficult for a network to fit the exact opposite effect of the other. As a result, it is more likely the model will decrease the effect of both irrelevant features during training. Third, since there are only a few intermediate variables, we can display the decision-making process and more easily discover these wrong decision logics.
>
> ***Comment 8**: A related work section should be added to address how this approach relates to other work in network explainability.*
>
> **Response**: Actually, we have a related work section in our paper (i.e., Section 4). Since VOTEN is a new model which not depends largely on the preliminary technologies, we have put the related work section right behind the experiment section.

---

### Decision · Program_Chairs · 2021-09-27

**Decision:**

Accept (Poster)

**Comment:**

This work proposes a novel network structure for better explainability. While there are many aspects of explainability that are not assessed in this paper, the local and global prediction processing cases are provided beside the accuracy. The term decision-making is a bit confusing as it often refers to decision-making tasks, while this paper handles prediction tasks. Overall, this paper is significantly novel, while its applicability is not quite clear.